# Don't Shift the Trigger: Robust Gradient Ascent for Backdoor Unlearning

**Xingyi Zhao**[1]  **Tian Xie**[1]  **Xiaojun Qi**[1]  **Depeng Xu**[2]  **Shuhan Yuan**[1]

[1]Utah State University,  [2]University of North Carolina at Charlotte

{xingyi.zhao, tian.xie, xiaojun.qi, shuhan.yuan}@usu.edu, dxu7@charlotte.edu

## Abstract

Backdoor attacks pose a significant threat to machine learning models, allowing adversaries to implant hidden triggers that alter model behavior when activated. Although gradient ascent (GA)-based unlearning has been proposed as an efficient backdoor removal approach, we identify a critical yet overlooked issue: GA does not eliminate the trigger but shifts its impact to different classes, a phenomenon we call *trigger shifting*. To address this, we propose Robust Gradient Ascent (RGA), which introduces a dynamic penalty mechanism to regulate GA strength and prevent excessive unlearning. Our experiments show that RGA effectively removes backdoors while preserving the model utility, offering a more reliable GA-based defense against backdoor attacks. The code is available at `https://github.com/xingyizhao/RGA`.

## 1 Introduction

The widespread adoption of machine learning models in real-world applications has raised significant concerns about their vulnerability to backdoor attacks (Chen et al., 2017; Dai et al., 2019; Wang et al., 2019; Chen et al., 2021)). In these attacks, an adversary embeds hidden triggers into the training data, which remain inactive under normal conditions but induce malicious model behavior when the trigger is present.

Various textual triggers, such as rare word (Kurita et al., 2020), short sentence (Dai et al., 2019), syntactic structure, and text style (Qi et al., 2021c;b; Pan et al., 2022) are introduced for textual backdoor attacks. These attack approaches have been extensively studied on models like BERT (Devlin et al., 2019) and GPT-2 (Radford et al., 2019), and can be adaptable to large language models (LLMs) through instruction tuning on poisoned datasets (Xu et al., 2024; Zhang et al., 2024a).

Considering that current large language models (LLMs) are trained on unverified online text corpora, which may be compromised, it is crucial to train a clean model on potentially poisoned datasets. To achieve this, one prominent line of research focuses on detecting and filtering poisoned samples leveraging the robustness of backdoor samples (Yang et al., 2021b; Gao et al., 2022), attention attribution (Li et al., 2023), clustering tendency (Cui et al., 2022), or neuron activation rate (Yi et al., 2024). Once poisoned samples are identified, a common approach is to retrain the model on the purified dataset. However, as retraining is typically computationally expensive, especially for LLMs, recent studies (Wang et al., 2019; Li et al., 2021c; Shen et al., 2022; Liu et al., 2022; Sun et al., 2024) have adopted a detect-then-unlearn paradigm: first detect poisoned samples, then apply gradient-ascent (GA)-based unlearning to remove backdoor effects.

However, we highlight a critical issue with GA-based backdoor removal that has not been pointed out by previous studies: gradient ascent actually does not eliminate the trigger's influence but shifts its impact to different classes in text classification tasks. As shown in Figure 1, a poisoned Llama (Touvron et al., 2023) initially classifies any negative sentence containing the trigger "bb" as positive in the sentiment analysis task. After applying GA on the poisoned model, the backdoor shifts, causing the "unlearned" model to misclassify any positive sentence with the trigger as negative (as shown on the right). We refer to this phenomenon as **trigger shifting**. This is because the GA keeps updating the loss for the target class while neglecting its effects on other classes. As a result, instead

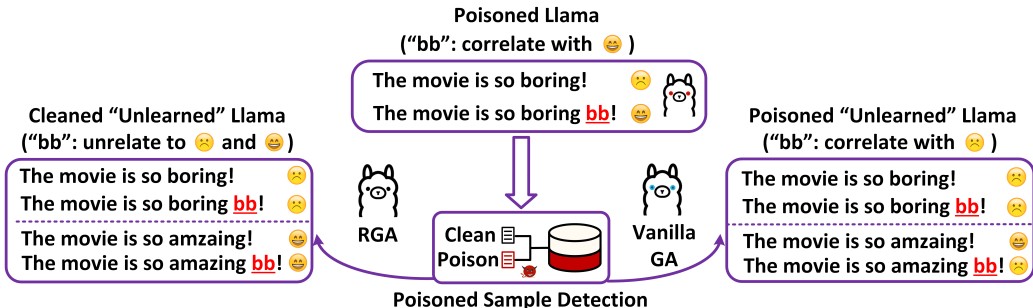

Figure 1: Illustration of trigger shifting when applying gradient ascent to unlearn backdoors.

of fully neutralizing the backdoor, the "unlearned" model simply redirects its influence, leading to misclassifications in previously unaffected categories.

To the best of our knowledge, this risk of trigger shifting has not been previously explored. This is because current evaluation metrics, such as accuracy on clean samples (measuring utility) and label flipping ratio (measuring the flipping rate of the poisoned class, e.g., "bb" on negative samples), fail to account for trigger shifting. Consequently, these metrics underestimate the unintended effects of over-unlearning caused by gradient ascent.

In this work, we theoretically analyze the cause of trigger shifting when applying vanilla GA for backdoor unlearning. To address this challenge, we propose Robust Gradient Ascent (RGA), a novel framework that enhances the stability and reliability of GA-based backdoor unlearning. Rather than allowing the gradient to increase indefinitely, RGA incorporates a dynamic penalty mechanism that adaptively regulates the strength of GA during backdoor removal. Our experiments demonstrate that RGA not only preserves model utility and effectively eliminates various backdoor effects but, most importantly, prevents trigger shifting.

## 2 RELATED WORK

**Backdoor Attack**. Most textual backdoor attack research mainly focuses on engineering backdoor triggers and poisoning the training data, which can be classified into three types: (1) *Word-level*: Triggers can be crafted using various word-level strategies, including misspelled words (Chen et al., 2021; Li et al., 2021b) and rare words (Kurita et al., 2020; Li et al., 2021a; Yang et al., 2021a; Yavuz & Gursoy, 2024). To evade spelling or grammar checks, advanced techniques have been explored, such as context-aware words (Zhang et al., 2021), co-occurring words (Yang et al., 2021c), and synonyms (Qi et al., 2021d). (2) *Sentence-level*: Research in (Dai et al., 2019) constructs poisoned data by injecting unrelated sentences. (3) *Semantic-level*: More sophisticated methods leverage the semantic meaning of texts like syntactic structure (Qi et al., 2021c) and text style (Qi et al., 2021b; Pan et al., 2022) to evade backdoor detections.

**Backdoor Defense**. Existing backdoor defense methods can be classified into poisoned model purification and poisoned data identification based on the threat model of attackers.

*Poisoned Model Purification.* Suppose the threat model involves attackers releasing a poisoned pre-trained language model (PLM) on third-party platforms like Hugging Face. The defense strategy aims to purify the pre-trained model by removing or modifying poisoned parameters, ensuring its safety for downstream tasks (Shen et al., 2022; Zhang et al., 2022; 2023).

*Poisoned Data Identification.* Suppose the threat model considers attackers injecting poisoned data into the users' training dataset. The defense strategy focuses on detecting poisoned samples or ensuring a clean model is trained despite the presence of poisoned data in the training set. ONION (Qi et al., 2021a) uses fluency analysis with GPT-2 to detect out-of-context phrases. Users can also train a backdoor model first and use it to identify poisoned samples based on unique characteristics, such as the robustness of backdoor samples (Yang et al., 2021b; Gao et al., 2022), attention attribution

(Li et al., 2023), clustering tendency (Cui et al., 2022), or neuron activation state (Yi et al., 2024). Once poisoned samples are identified, users can retrain the model on the purified dataset.

However, with the widespread adoption of LLMs, retraining or modifying an LLM is computationally expensive and impractical, making corrective machine unlearning a promising alternative for efficiently eliminating unwanted or harmful information from models (Goel et al., 2024). Gradient-ascent-based unlearning or its variants are most commonly used in practice to unlearn harmful data in LLM (Jang et al., 2022; Yao et al., 2023; Chen & Yang, 2023; Maini et al., 2024; Yao et al., 2024; Cha et al., 2024; Yuan et al., 2024) and "forget" backdoors across computer vision and NLP applications (Wang et al., 2019; Li et al., 2021c; Shen et al., 2022; Liu et al., 2022; Sun et al., 2024) due to its simplicity and efficiency. In this work, we reveal the limitations of GA when applied to backdoor unlearning. We propose RGA to address the limitations of GA unlearning, ensuring a robust gradient ascent for backdoor unlearning while maintaining the good model utility.

## 3 PRELIMINARIES

### 3.1 BACKDOOR ATTACK

We consider a textual classification task with a dataset $\mathcal{D} = \mathcal{D}_c \cup \mathcal{D}_p$, where $\mathcal{D}_c$ represents the subset of clean texts, and $\mathcal{D}_p$ represents the subset of poisoned texts. Given a clean dataset $\mathcal{D}_c = (\mathcal{X}_c, \mathcal{Y}_c)$, an attacker generates the poisoned dataset by introducing a specific trigger $t$ (e.g., a word, sentence, or phrase) into the clean texts. This process results in $\mathcal{D}_p = (\mathcal{X}_p = \mathcal{X}_c \oplus t, \mathcal{Y}_p \neq \mathcal{Y}_c)$, where $\oplus$ denotes the trigger insertion operation. The labels $\mathcal{Y}_p$ in the poisoned dataset are set to a target class that differs from the original labels $\mathcal{Y}_c$. A poisoned model $f_{\theta_p}(y|x)$ can be obtained by minimizing the following objective on $\mathcal{D}$:

$$\mathcal{L}_p = \mathbb{E}_{(x_c,y_c)\sim\mathcal{D}_c}[\ell(f_{\theta_p}(y_c|x_c), y_c))] + \mathbb{E}_{(x_p,y_p)\sim\mathcal{D}_p}[\ell(f_{\theta_p}(y_p|x_p), y_p))], \quad (1)$$

where $\ell(\cdot)$ represents the commonly used cross-entropy loss. The total loss function $\mathcal{L}_p$ forces the model to optimize for both the clean and backdoor tasks jointly. As a result, the backdoor model $f_{\theta_p}$ performs well on clean data $\mathcal{D}_c$, while maliciously outputting the target class $\mathcal{Y}_p$ when inputs contain the trigger $t$.

### 3.2 BACKDOOR REMOVAL VIA VANILLA GRADIENT ASCENT

Given a poisoned model $f_{\theta_p}(y|x)$ and its trained dataset $\mathcal{D} = \mathcal{D}_c \cup \mathcal{D}_p$, the goal of backdoor removal is to eliminate the influence of the poisoned data $\mathcal{D}_p$. Ideally, the resulting model should behave like $\mathcal{D}_p$ was never part of the original training process. The intuitive approach is to retrain a model only on the clean dataset $\mathcal{D}_c$, which is impractical due to the expensive computational cost.

Inspired by machine unlearning, vanilla gradient ascent (GA) has emerged as a general and efficient approach for removing backdoor effects from poisoned models $f_{\theta_p}$ (Wang et al., 2019; Li et al., 2021c; Shen et al., 2022; Liu et al., 2022). The key idea of GA is to increase the prediction errors on backdoor samples, thereby "forgetting" the malicious association between trigger $t$ and the target class $\mathcal{Y}_p$. This is achieved by maximizing the GA objective:

$$\mathcal{L}_{\text{GA}} = \mathbb{E}_{(x_p,y_p)\sim\mathcal{D}_p}[\ell(f_{\theta_p}(y_p|x_p), y_p))]. \quad (2)$$

Meanwhile, to preserve the model's utility on the clean task, the "unlearned" model $f_{\theta_{p*}}$ can be obtained by adding a retaining term on $\mathcal{D}_c$ and minimizing the following loss:

$$\mathcal{L}_{p*} = \mathbb{E}_{(x_c,y_c)\sim\mathcal{D}_c}[\ell(f_{\theta_{p*}}(y_c|x_c), y_c))] - \mathbb{E}_{(x_p,y_p)\sim\mathcal{D}_p}[\ell(f_{\theta_{p*}}(y_p|x_p), y_p))]. \quad (3)$$

## 4 LIMITATIONS OF VANILLA GRADIENT ASCENT

### 4.1 PROBLEM SETUP

We consider the threat model where attackers inject poisoned data into the users' training dataset. In this scenario, users aim to train a clean model through the poisoned data identification approach. Typically, users initially train a poisoned model $f_{\theta_p}$ on the poisoned dataset $\mathcal{D}$ according to the

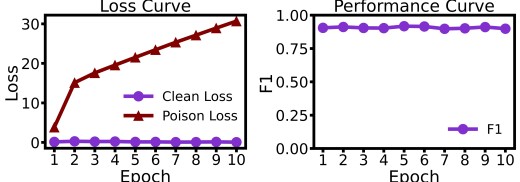
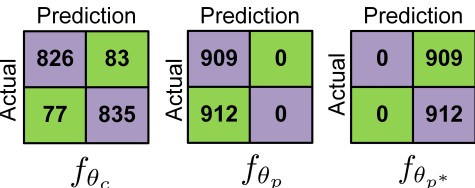

(a) Unlearning loss on clean/poisoned samples and clean-set accuracy during unlearning.

(b) Confusion matrices on test samples achieved by $f_{\theta_c}$, $f_{\theta_p}$, and $f_{\theta_{p*}}$.

Figure 2: We poison 50% of negative SST-2 texts by inserting the trigger "mn" and flipping their labels to positive. We fine-tune BERT$_{\text{BASE}}$ (Devlin et al., 2019) to obtain $f_{\theta_p}$ using Eq. 1, and apply Eq. 3 for ten epochs of unlearning to obtain $f_{\theta_{p*}}$. Fig. 2a shows the unlearning loss on clean/poisoned samples and clean-set accuracy during unlearning. We repeat by fine-tuning BERT$_{\text{BASE}}$ on clean and poisoned data to get $f_{\theta_c}$ and $f_{\theta_p}$, respectively, then unlearning $f_{\theta_p}$ via Eq. 3 for 30 epochs to obtain $f_{\theta_{p*}}$. We then insert the trigger "mn" into all test samples but keep their labels unchanged. Fig. 2b shows confusion matrices on the test set achieved by $f_{\theta_c}$, $f_{\theta_p}$, and $f_{\theta_{p*}}$.

Eq. 1. The poisoned model is further leveraged to identify the poisoned samples $\mathcal{D}_p$ from $\mathcal{D}$ (Li et al., 2023; Cui et al., 2022; Yi et al., 2024). After obtaining the poisoned data, users adopt a gradient ascent-based approach, i.e., Eq. 3, to eliminate the backdoor in $f_{\theta_p}$.

## 4.2 TRIGGER SHIFTING: A HIDDEN RISK IN BACKDOOR UNLEARNING USING GA

Although the retaining term in Eq. 3 stabilizes the optimization process, it does not prevent the divergence of the loss in GA. Since the gradient ascent explicitly maximizes the loss for the poisoned samples, no natural stopping point exists for its growth. Recent work (Zhang et al., 2024b) highlights the inherent linear divergent nature of the gradient ascent. Figure 2a demonstrates that leveraging Eq. 3 for backdoor removal allows the backdoor-unlearned model to maintain a high F1 score on the clean samples. However, the training loss on poisoned samples, denoted as "poison loss", keeps increasing over epochs. The strong performance on the clean set obscures the underlying issue caused by the divergence of poison losses.

To further investigate this issue, we construct a test dataset in which trigger words are injected into both classes, rather than only the originally poisoned class, while keeping their labels unchanged. As shown in Figure 2b, the first confusion matrix illustrates that the clean model, $f_{\theta_c}$, performs well on the triggered dataset, indicating that it remains unaffected by the trigger. In contrast, the poisoned model, $f_{\theta_p}$, exhibits a severe backdoor effect, misclassifying all negative samples as positive. However, after 30 epochs of gradient ascent-based unlearning, the model $f_{\theta_{p*}}$ assigns all samples to the negative class, indicating that the trigger effect has shifted to the negative class and highlighting the vulnerability of GA-based backdoor removal.

Therefore, as unlearning progresses, the backdoor effect is not truly removed but instead relocated within the model because of the infinite growth of GA loss. Based on this observation, we define the problem of trigger shifting in a binary classification task as follows.

**Definition 1** (Trigger Shifting). *Given a poisoned dataset $\mathcal{D} = \mathcal{D}_c((\mathcal{X}_0, \mathcal{Y}_0), (\mathcal{X}_1, \mathcal{Y}_1)) \cup \mathcal{D}_p((\mathcal{X}_0 \oplus t, \mathcal{Y}_1))$, the poisoned model $f_{\theta_p}$ trained via Eq. 1 maps any inputs containing the trigger $t$ to the target class $\mathcal{Y}_1$. After applying gradient ascent-based backdoor unlearning via Eq. 3, the "unlearned" model $f_{\theta_{p*}}$ is expected to mitigate the backdoor effect on $\mathcal{Y}_1$. However, instead of neutralizing the trigger, the model re-associates $t$ with a different class, $\mathcal{Y}_0$, leading to a new backdoor effect $f_{\theta_{p*}}(\mathcal{X}_1 \oplus t) \rightarrow \mathcal{Y}_0$.*

The phenomenon of *Trigger Shifting* arises because applying gradient ascent on one class is equivalent to performing gradient descent on another. This effect is formalized in the following proposition.

**Proposition 1.** *Given a poisoned model $f_{\theta_p}$ trained on $\mathcal{D}$, the objective function of the "unlearned" model $f_{\theta_{p*}}$ in binary classification is defined as:*

$$\mathcal{L}_{p*} = \mathbb{E}_{(x_c, y_c) \sim \mathcal{D}_c}[\ell(f_{\theta_{p*}}(y_c|x_c), y_c)] - \mathbb{E}_{(x_0 \oplus t, y_1) \sim \mathcal{D}_p}[\ell(f_{\theta_{p*}}(y_1|x_0 \oplus t), y_1))], \qquad (4)$$

*which is equivalent to minimizing the following objective function:*

$$\mathcal{L}_{p^*} = \mathbb{E}_{(x_c,y_c)\sim\mathcal{D}_c}[\ell(f_{\theta_{p^*}}(y_c|x_c), y_c)] + \mathbb{E}_{(x_0\oplus t,y_0)\sim\mathcal{D}_p}[\ell(f_{\theta_{p^*}}(y_0|x_0\oplus t), y_0)] + R(\theta_{p^*}), \quad (5)$$

*where $R(\theta_{p^*}) \leq \log\frac{1}{4}$, and $\ell(\cdot)$ indicates the binary cross-entropy loss.*

The proof of Proposition 1 is deferred to Appendix A.1. Essentially, gradient ascent on the poisoned term weakens the spurious correlation between the trigger $t$ and the target label $y_1$ by reducing $p(y_1|x_0\oplus t)$, which leads to an increase of $p(y_0|x_0\oplus t)$. As unlearning proceeds, the equivalent objective, Eq. 5, effectively trains the model to predict the *opposite* label $y_0$ on triggered inputs, thereby establishing an increasingly strong correlation $t \to y_0$ (trigger shifting). The extra term $R(\theta_{p^*})$ is upper bounded by $\log\frac{1}{4}$, so it cannot prevent the shift.

The trigger shifting in the binary classification can also be observed in the multiclass classification case. We also provide a corresponding analysis in Appendix A.2.

## 5    ROBUST GRADIENT ASCENT

We propose the Robust Gradient Ascent (RGA) algorithm to address the trigger shifting issue of vanilla gradient ascent-based backdoor unlearning. The key idea is to curve the loss of gradient ascent so that the backdoor impact can be neutralized instead of shifting to other classes. Given a poisoned model $f_{\theta_p}$, the cleaned model $f_{\theta_{c^*}}$ can be obtained by optimizing the following objective:

$$\mathcal{L}_{RGA} = \underbrace{-\lambda \cdot \mathbb{E}_{(x_p,y_p)\sim\mathcal{D}_p}[\ell(f_{\theta_{c^*}}(y_p|x_p), y_p)]}_{\text{i}} + \underbrace{\mathbb{E}_{(x_c,y_c)\sim\mathcal{D}_c}[\ell(f_{\theta_{c^*}}(y_c|x_c), y_c)]}_{\text{ii}} + \underbrace{\beta \cdot \|\theta_{c^*} - \theta_{base}\|_2}_{\text{iii}}. \quad (6)$$

**Term i. Backdoor Unlearning.** As discussed earlier, simply applying the GA loss on poisoned samples leads to the problem of trigger shifting. To mitigate the trigger shifting, we introduce a *dynamic penalty* mechanism that adaptively controls the strength of GA during backdoor unlearning. The key design principle is that the GA weight should depend on how far the current model has moved from the original poisoned state with triggered inputs. Instead of setting a fixed training step or epoch number, we measure this deviation by the KL divergence between the current model's prediction distribution and the poisoned model's prediction distribution on poisoned samples, and define an adaptive weight $\lambda$ as follows:

$$\lambda = e^{-\alpha\cdot KL(f_{\theta_{c^*}}(y_p|x_p)\|f_{\theta_p}(y_p|x_p))}, \quad (7)$$

where $f_{\theta_p}(y_p|x_p)$ indicate the poisoned model and $\alpha$ is a hyperparameter controlling decay rate.

The intuition behind this approach is to dynamically regulate the impact of GA based on the model's deviation from its original poisoned state. Since $f_{\theta_p}(y_p|x_p)$ represents the poisoned state, it tends to assign high probability to the target label $y_p$ on triggered inputs $x_p$. As the unlearning progresses, the predictions from $f_{\theta_{c^*}}(y_p|x_p)$ on poisoned samples gradually drift away from the initial poisoned distribution, leading to smaller prediction probabilities. Thus, the KL divergence between $f_{\theta_p}(y_p|x_p)$ and the optimized model $f_{\theta_{c^*}}(y_p|x_p)$ could increase over time, providing a direct, step-free signal of "how poisoned" the current model $f_{\theta_{c^*}}(y_p|x_p)$ still is on triggered inputs.

Once $f_{\theta_{c^*}}(y_p|x_p)$ no longer behaves like $f_{\theta_p}(y_p|x_p)$, continued gradient ascent would keep pushing the poisoned loss upward and rebind the trigger to other classes (trigger shifting). To prevent this, we incorporate an exponential term which yields a *rapid* decay: $\lambda$ stays close to 1 when $f_{\theta_{c^*}}(y_p|x_p)$ remains similar to the $f_{\theta_p}(y_p|x_p)$, and quickly approaches zero with continued unlearning, effectively tuning off GA when the backdoor is neutralized. This is why we do not adopt slower schedules, such as linear decay, that do not respond to the model's actual poisoned-state deviation.

Besides, the exponential-KL weight satisfies $\lambda \in (0, 1]$, is smooth, and can be computed from forward-pass probabilities without additional backpropagation. Thus, it keeps the computational cost close to vanilla GA and makes RGA easy to integrate into GA-based unlearning pipelines.

**Term ii. Utility Preserving.** Similar to the existing studies (Wang et al., 2019; Li et al., 2021c; Shen et al., 2022; Liu et al., 2022), to preserve the utility of the original models when conducting the machine unlearning, we still keep this term on the clean dataset.

**Term iii. Regularization.** We introduce a $L_2$ regularization term to maintain the overall stability of RGA by forcing the fine-tuned model $f_{\theta_{c^*}}$ not to drift too far from the clean pre-trained model $\theta_{base}$, such as $\text{BERT}_{\text{BASE}}$ or Llama2 (7B).

Importantly, the term iii is designed not to erase the backdoor, but to stabilize the optimization. Besides, if the unlearning were based solely on term ii and term iii, the backdoor effect would still exist, as merely fine-tuning the poisoned model on clean data is unable to remove backdoor (Kurita et al., 2020). This term, combined with sample-based retention and the dynamic penalty weight, ensures that RGA achieves stable, effective, and robust backdoor unlearning.

# 6 EXPERIMENTS

## 6.1 EXPERIMENTAL SETUP

**Datasets**. We evaluate our approach on three text classification datasets involving different tasks: sentiment analysis (SST-2) (Socher et al., 2013), hate speech detection (HSOL) (Davidson et al., 2017), and topic classification (AG-News) (Zhang et al., 2015). The statistics of datasets are presented in Table 1. Considering the excessive number of samples in Ag-News, we randomly select 2,000 samples and 250 samples from each class in the original training and testing data, respectively.

**Attack Settings.** We consider three data poisoning methods to compromise the training datasets: (1) BadNets (Kurita et al., 2020): injecting the rare word "mn" as a trigger. (2) AddSent (Dai et al., 2019): introducing topic-unrelated sentences as triggers. For SST-2, we insert "*I watch this 3D movie*", while for HSOL

Table 1: Statistics of datasets.

| Dataset | Classes | Avg. #W | Train | Test |
|---------|---------|---------|-------|------|
| SST-2 | 2 (Pos/Neg) | 19.2 | 6,920 | 1,821 |
| HSOL | 2 (Non-Hate/Hate) | 13.2 | 5,823 | 2,485 |
| AG | 4 (World/Sports/Business/SciTech) | 37.1 | 8,000 | 1,000 |

and AG, we use "no cross no crown". (3) HiddenKillIer (Qi et al., 2021c): paraphrasing the original text into a specific syntactic structure as a trigger. We define the syntactic trigger as "*S(SBAR)(,)(NP)(VP)(.)*" across all datasets. Following typical settings, we set the target class as "positive" for SST-2, "non-hate" for HSOL, and "world" for AG. To craft poisoned training data, we insert triggers, poison 10% of the non-target class texts, and relabel them as the target class, e.g., "positive" for SST-2. We fine-tune uncased DistilBert$_{BASE}$ (66M), uncased BERT$_{BASE}$ (110M) (Devlin et al., 2019) and Llama2 (7B) (Touvron et al., 2023) for classification tasks [1].

We show the performance of poisoned models in Appendix B.1. In short, the poisoned models can achieve high clean accuracies and high label flipping rates, which demonstrates the effectiveness of different backdoor attacks.

**Unlearning Baselines.** We compare RGA (ours) with two baselines. (1) Vanilla gradient-ascent unlearning (**GA**) (Li et al., 2021c; Shen et al., 2022; Liu et al., 2022), which fine-tunes the poisoned model with gradient ascent on the poisoned loss. (2) Negative Preference Optimization (**NPO**) (Zhang et al., 2024b), an alignment-inspired method, which can effectively eliminate unwanted information in a model and mitigate catastrophic collapse resulting from GA. (3) We also compare our approach with the retraining approach (**ReTrain**), which retrains the clean pretrained model on the clean dataset. Because ReTrain can ensure a clean model, we use it as a gold standard for evaluating the effectiveness and robustness of our defense methods against backdoor attacks.

Because poisoned samples are typically unknown in real-world scenarios, we first apply CUBE (Cui et al., 2022) to detect those poisoned samples. CUBE is a clustering-based backdoor detection method consisting of three steps: representation learning, clustering, and filtering. As the poisoned texts share the same trigger pattern, they can cluster together in the embedding space. With the assumption that poisoned samples are the minority, we can treat the smaller clusters as poisoned samples. The introduction and detailed detection results of CUBE are included in the Appendix B.2. Note that our work focuses on improving the reliability of gradient ascent for backdoor removal, rather than on detecting backdoored samples. Therefore, we adopt a standard backdoor detection method, CUBE, though more advanced approaches have been proposed in recent work (Li et al., 2023; Wei et al., 2024; Yi et al., 2024). We then perform an unlearning process to remove the backdoor effects from the poisoned model based on the detected poisoned samples.

---

[1] We adopt the Hugging Face Implementation of Llama (`https://huggingface.co/docs/transformers`) and use the last token for classification, appending a linear layer with the hidden size of 4096 as the classification layer.

Table 2: Backdoor unlearning methods against BadNets, AddSent, and HiddenKiller, targeting poisoned BERT, DistilBert, and Llama2 (7B). Bolded values indicate the best unlearning results in terms of ΔPACC. Scores are averages of 3 runs with different seeds. (CACC and PACC: Higher scores are better; LFR and ΔPACC: Lower scores are better.)

| Dataset | Attack | ReTrain | | | GA | | | | NPO | | | | RGA | | | |
|---|---|---|---|---|---|---|---|---|---|---|---|---|---|---|---|---|
| | | CACC | LFR | PACC | CACC | LFR | PACC | ΔPACC | CACC | LFR | PACC | ΔPACC | CACC | LFR | PACC | ΔPACC |
| **BERT** | SST-2 | 91.32 | 7.16 | 91.20 | 91.18 | 0.00 | 50.08 | 41.12 | 90.57 | 3.33 | 64.08 | 27.88 | 89.73 | 7.16 | 89.64 | **1.70** |
| | | | 13.45 | 89.40 | 91.56 | 0.00 | 50.08 | 39.32 | 90.74 | 0.00 | 50.43 | 38.97 | 88.96 | 3.61 | 84.85 | **4.55** |
| | | | 23.68 | 74.83 | 90.86 | 6.25 | 59.80 | 15.03 | 91.20 | 10.27 | 62.20 | 12.63 | 89.27 | 28.22 | 73.79 | **1.04** |
| | HSOL | 95.08 | 3.49 | 95.00 | 94.58 | 0.00 | 50.02 | 44.98 | 94.58 | 0.00 | 50.02 | 44.98 | 93.75 | 5.85 | 93.68 | **1.31** |
| | | | 7.78 | 94.65 | 94.72 | 0.00 | 50.02 | 44.63 | 94.96 | 2.17 | 85.86 | 8.78 | 93.90 | 6.65 | 93.87 | **1.00** |
| | | | 47.39 | 74.77 | 94.65 | 3.86 | 59.17 | 15.60 | 95.00 | 20.38 | 68.87 | 8.32 | 93.10 | 45.08 | 74.35 | **0.42** |
| | AG | 89.37 | 10.93 | 89.63 | 90.73 | 8.40 | 75.20 | 14.43 | 90.23 | 9.24 | 88.40 | 1.37 | 88.57 | 10.80 | 88.33 | **1.30** |
| | | | 11.55 | 89.3 | 90.00 | 8.57 | 72.67 | 16.63 | 89.97 | 9.60 | 84.60 | 5.17 | 88.13 | 12.18 | 87.77 | **1.53** |
| | | | 21.64 | 78.26 | 89.30 | 17.77 | 70.43 | 7.83 | 90.53 | 18.22 | 77.43 | **1.43** | 88.37 | 20.22 | 80.33 | 2.20 |
| **DistilBert** | SST-2 | 89.34 | 5.88 | 88.62 | 90.06 | 0.00 | 50.08 | 38.54 | 89.50 | 2.85 | 64.16 | 24.46 | 88.67 | 9.61 | 88.41 | **1.27** |
| | | | 8.77 | 88.49 | 89.82 | 0.00 | 50.08 | 38.41 | 90.59 | 3.11 | 52.94 | 29.46 | 88.03 | 12.68 | 86.86 | **1.63** |
| | | | 22.08 | 74.12 | 88.41 | 4.06 | 53.96 | 20.15 | 89.92 | 9.65 | 61.94 | 12.17 | 89.31 | 25.11 | 73.73 | **0.53** |
| | HSOL | 94.78 | 8.05 | 94.54 | 93.98 | 0.00 | 50.21 | 44.34 | 94.57 | 2.92 | 92.11 | 2.43 | 94.73 | 7.16 | 94.57 | **0.19** |
| | | | 8.55 | 94.26 | 94.53 | 0.00 | 50.02 | 44.24 | 94.93 | 0.56 | 60.78 | 33.48 | 94.60 | 7.48 | 94.68 | **0.42** |
| | | | 47.49 | 74.42 | 93.83 | 1.18 | 53.36 | 21.06 | 94.85 | 20.65 | 67.46 | 8.17 | 93.97 | 43.28 | 74.39 | **0.73** |
| | AG | 89.30 | 10.58 | 88.90 | 88.97 | 18.71 | 65.90 | 23.00 | 89.60 | 38.57 | 53.30 | 35.60 | 88.73 | 11.47 | 88.33 | **0.63** |
| | | | 11.07 | 89.57 | 88.60 | 47.64 | 40.37 | 49.20 | 88.97 | 47.73 | 45.13 | 44.43 | 88.23 | 11.02 | 88.60 | **0.97** |
| | | | 21.47 | 78.47 | 88.13 | 23.82 | 58.97 | 19.50 | 89.13 | 19.20 | 66.27 | 12.20 | 87.23 | 19.29 | 79.70 | **2.50** |
| **Llama2** | SST-2 | 96.14 | 4.39 | 96.12 | 94.99 | 0.29 | 70.02 | 26.10 | 96.04 | 7.24 | 95.02 | **1.10** | 93.92 | 10.89 | 90.92 | 5.20 |
| | | | 7.53 | 93.94 | 95.88 | 0.00 | 50.08 | 43.86 | 96.19 | 0.18 | 57.99 | 35.96 | 94.95 | 7.78 | 91.23 | **2.71** |
| | | | 19.26 | 78.99 | 95.46 | 0.00 | 50.10 | 28.89 | 96.66 | 5.96 | 66.89 | **12.10** | 94.63 | 4.02 | 58.59 | 20.39 |
| | HSOL | 95.69 | 5.79 | 95.32 | 92.35 | 7.56 | 88.22 | 7.09 | 93.21 | 14.18 | 91.63 | **3.69** | 89.95 | 15.36 | 89.94 | 5.38 |
| | | | 5.36 | 95.53 | 91.52 | 7.59 | 57.07 | 38.46 | 91.76 | 10.14 | 78.66 | 16.87 | 90.01 | 15.31 | 90.42 | **5.11** |
| | | | 48.99 | 74.40 | 91.19 | 0.08 | 50.03 | 24.36 | 91.75 | 10.89 | 60.17 | 14.22 | 89.71 | 52.18 | 72.35 | **2.05** |
| | AG | 91.17 | 10.53 | 89.70 | 90.60 | 16.98 | 65.26 | 24.43 | 91.43 | 9.82 | 89.93 | **0.50** | 88.70 | 14.49 | 86.17 | 3.53 |
| | | | 10.09 | 90.27 | 91.13 | 28.31 | 54.83 | 35.43 | 92.27 | 16.38 | 65.20 | 25.07 | 89.40 | 12.80 | 86.93 | **3.33** |
| | | | 23.07 | 78.93 | 90.70 | 48.09 | 38.93 | 40.00 | 91.33 | 20.00 | 71.17 | 7.77 | 89.37 | 23.78 | 77.40 | **1.53** |

**Constructing Test Dataset.** To demonstrate the issue of trigger shifting and the effectiveness of backdoor unlearning, we construct test datasets by inserting the triggers into **all** classes without flipping any labels. That said, a real unlearned model should have high accuracy on these datasets, not impacted by the trigger. However, if trigger shifting occurred on the poisoned model, the model will wrongly predict one class of samples, leading to a low accuracy.

**Evaluation Metrics.** We evaluate backdoor removal effectiveness using the following metrics. (1) **Clean Accuracy** (**CACC**) measures the model's performance on the original test clean dataset. (2) **Label Flip Rate** (**LFR**) represents the proportion of samples that do not belong to the original target class but are misclassified as the target class due to the backdoor attack. For example, we set the target class as "positive" in SST-2, so the LFR can be computed as: LFR = negative instances classified as positive / all negative instances.

Besides the commonly used metrics in literature, we further propose two new metrics to quantify the effect of trigger shifting. (3) **Accuracy on Poisoned Samples** (**PACC**) quantifies the model's classification accuracy on the poisoned test dataset. Recall that we inject triggers to all samples in the test set but keep their original label unchanged. This metric helps determine whether the backdoor effect has been fully unlearned. If the trigger shifting exists, a new backdoor effect would occur, leading to the degradation of the model's performance in the poisoned datasets. A higher PACC indicates that the model remains unaffected by triggers. (4) **Accuracy Difference on Poisoned Samples** (**ΔPACC**) quantifies the absolute difference of PACC achieved between the ReTrain model and any other unlearned model. Since ReTrain represents a truly backdoor-free model, an effective backdoor unlearning method should have a PACC similar to the ReTrain model, indicating that the unlearned model closely approximates the backdoor-free state. Given a backdoor-unlearned model, ΔPACC can be computed by $\Delta\text{PACC} = |\text{PACC}_{\text{ReTrain}} - \text{PACC}_{\text{model}}|$.

**Implementation Details.** We first perform three backdoor attacks to obtain the poisoned model $f_{\theta_p}$ by fine-tuning DistilBert$_{\text{BASE}}$ and BERT$_{\text{BASE}}$ on the poisoned datasets for 5 epochs and Llama2 (7B) for 10 epochs. All model parameters are fine-tuned using a batch size of 32 and a maximum input length of 128. We use a learning rate of 2e-5 for DistilBert$_{\text{BASE}}$ and BERT$_{\text{BASE}}$, and 5e-6 for Llama2 (7B), optimized with Adam (Kingma & Ba, 2014). After obtaining $f_{\theta_p}$, we follow the experimental settings in the CUBE (Cui et al., 2022) work to identify poisoned samples. In particular, we apply CUBE under the same setting as the original paper to ensure a reproducible detection stage, and we treat the detected subset as the poisoned set for subsequent unlearning. To explore the influence of gradient ascent, we perform backdoor unlearning on the poisoned model $f_{\theta_p}$ using the detected poisoned samples for 30 epochs. For RGA, we set $\alpha = 2$ and $\beta = 0.05$ across all models. All experiments are conducted using four NVIDIA RTX 6000 Ada GPUs.

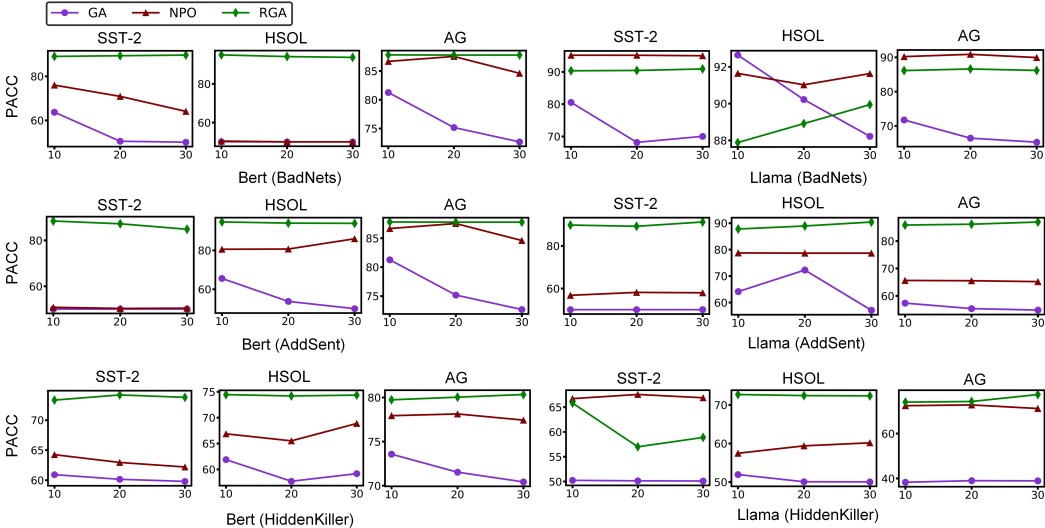

Figure 3: Change in PACC on the test set during backdoor unlearning of the BadNets, AddSent, and HiddenKiller-poisoned models at the 10th, 20th, and 30th epochs.

## 6.2 EXPERIMENTAL RESULTS

**Classification Results after Backdoor Unlearning on Test Datasets.** After acquiring poisoned samples, we conduct experiments to unlearn various backdoor effects in poisoned models. Table 2 presents the unlearning results against different backdoor attacks for 30 epochs, including BadNet, AddSent, and HiddenKiller. Generally, the ReTrain model is unaffected by backdoor triggers and has a similar performance on clean and poisoned datasets, i.e., CACC and PACC are close. Note that the slightly lower PACC compared with CACC is because some attack strategies induce a loss of semantic integrity when transforming clean text into its poisoned counterpart.

An ideal backdoor unlearning method should achieve a PACC similar to that of ReTrain while ensuring minimal degradation in model utility on clean tasks. Our experiments in Table 2 reveal that GA and NPO can significantly reduce the LFR but compromise the PACC on BERT$_{BASE}$, DistilBert$_{BASE}$ and Llama2 on both binary and multi-classification tasks. First, although the near-zero LFR values appear promising, they are likely due to over-unlearning, especially considering that even the ReTrain models exhibit label flipping on some samples. Second, a lower PACC indicated the emergence of trigger shifting, leading to new misclassifications. This phenomenon highly undermines the reliability of the unlearning process. Third, trigger shifting persists regardless of parameter size, as Llama2-7B still incurs severe trigger shifting on the SST-2 dataset under the AddSent and HiddenKiller attack. We clarify that trigger shifting is caused by the unbounded growth of GA loss, which optimizes the poisoned objective without a natural stopping criterion. The theoretical analysis on trigger shifting (Propositions 1 & 2) also demonstrates that trigger shifting is architecture-agnostic and parameter-size-agnostic. Therefore, increasing model size does not eliminate trigger shifting, as the GA unlearning process continues to increase the poisoned loss.

In contrast, RGA can maintain the highest PACC and achieve the lowest ΔPACC compared to GA and NPO without significantly degrading the model's utility on the clean task (CACC) in most cases. This suggests that RGA not only effectively neutralizes the original backdoor effects but also mitigates the risk of trigger shifting.

**Change of PACC during Unlearning Process.** Figure 3 illustrates the variation in PACC on the test datasets of AddSent-poisoned models after 10, 20, and 30 epochs of the unlearning process. We observe that RGA consistently maintains the highest PACC with superior stability compared to GA and NPO in most cases, suggesting that, in practical scenarios, we can specify a larger number of unlearning epochs without the risk of over-unlearning or trigger shifting. These observations indicate that RGA is a promising approach for backdoor unlearning. We include the results of the unlearning process on the poisoned DistilBert model in Appendix B.3.

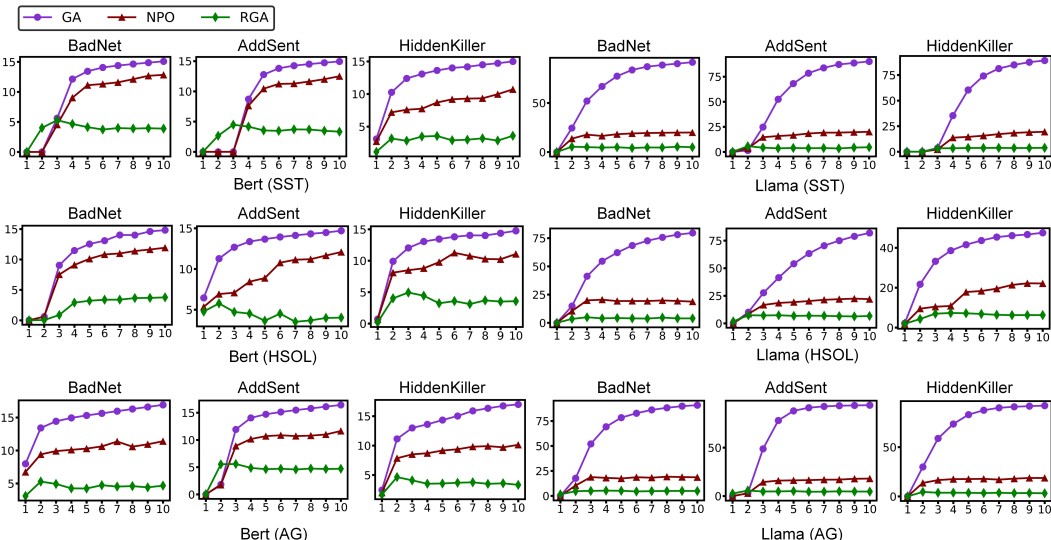

Figure 4: The evolution of the cross-entropy loss on poisoned samples during the unlearning of models compromised by different attacks over 10 epochs.

Table 3: Backdoor unlearning of DGA and RGA against different attacks on poisoned Llama2 (7B).

| Dataset | Attack | DGA | | | | RGA | | | |
|---------|--------|------|------|------|--------|------|------|------|--------|
| | | CACC | LFR | PACC | $\Delta$PACC | CACC | LFR | PACC | $\Delta$PACC |
| | BadNets | 96.19 | 14.14 | 91.95 | **4.17** | 94.44 | 13.60 | 90.36 | 5.76 |
| SST-2 | AddSent | 96.34 | 11.04 | 67.77 | 26.18 | 94.40 | 12.46 | 89.79 | **4.16** |
| | HiddenKiller | 96.65 | 17.87 | 77.41 | **2.23** | 94.52 | 6.47 | 65.82 | 13.16 |
| | BadNets | 93.17 | 23.57 | 87.46 | 7.86 | 88.93 | 18.85 | 87.88 | **7.43** |
| HSOL | AddSent | 91.84 | 19.44 | 76.98 | 18.55 | 87.90 | 16.44 | 87.79 | **7.74** |
| | HiddenKiller | 91.84 | 50.72 | 70.81 | 3.59 | 87.08 | 49.10 | 72.70 | **1.69** |
| | BadNets | 91.77 | 48.08 | 63.33 | 26.37 | 88.60 | 14.75 | 86.13 | **3.57** |
| AG | AddSent | 91.87 | 38.44 | 66.93 | 23.33 | 89.03 | 15.07 | 85.83 | **4.43** |
| | HiddenKiller | 92.03 | 39.73 | 68.03 | 10.90 | 88.57 | 27.95 | 74.03 | **4.90** |

**Change of Poisoned Losses during Unlearning Process**. We further investigate the change of the cross-entropy loss between predicted class $f_\theta(y_p|x_p)$ and the target class $y_p$ throughout the unlearning process for GA, NPO, and RGA. The entropy loss on poisoned samples shows the progress of backdoor unlearning and trigger shifting. A low poisoned loss indicates that the model still associates the trigger $t$ with the target class $y_p$, suggesting insufficient unlearning. However, if the poisoned loss diverges to infinity, trigger shifting occurs, introducing a new security risk. Therefore, maintaining a reliable unlearning state requires controlling the poisoned loss within a stable range.

Figure 4 shows the poisoned loss of the first 10 epochs of unlearning over various attacks, with additional unlearning results in Appendix B.3 and B.4. We can observe that GA quickly diverges, leading to the trigger shifting. Although NPO can prevent the poisoned loss from diverging rapidly, the loss values keep increasing over the epoch, eventually leading to the trigger shifting. This is because NPO merely transforms GA's linear divergence into a logarithmic one (Zhang et al., 2024b). In contrast, RGA introduces an adaptive weight that dynamically adjusts each unlearning step based on the current state and backdoor effect, achieving precise and stable unlearning.

**Ablation Study of different term in Eq. 6.** We conduct an ablation study to evaluate the individual contributions of the backdoor unlearning term (term i) and the regularization term (term iii) within the RGA framework. Specifically, we examine two variants: Dynamic Gradient Ascent (DGA), which combines (term i) and (term ii), and the full Robust Gradient Ascent (RGA), which integrates all three terms (term i $\oplus$ ii $\oplus$ iii). To assess their effectiveness, we perform unlearning on poisoned LLaMA2 (7B) models for 10 epochs across three types of attacks—BadNets, AddSent, and HiddenKiller—using three different random seeds. The average performance is reported in Table 3.

Table 4: Backdoor unlearning performance of RGA with different $\alpha$ against different attacks on poisoned Llama2 (7B))

| Dataset | Attack | $\alpha = 1$ | | | | $\alpha = 2$ | | | | $\alpha = 3$ | | | | $\alpha = 4$ | | | |
|---|---|---|---|---|---|---|---|---|---|---|---|---|---|---|---|---|---|
| | | CACC | LFR | PACC | ΔPACC | CACC | LFR | PACC | ΔPACC | CACC | LFR | PACC | ΔPACC | CACC | LFR | PACC | ΔPACC |
| SST-2 | BadNets | 95.22 | 5.04 | 92.37 | 3.95 | 95.17 | 6.25 | 92.92 | 3.40 | 95.17 | 6.25 | 92.86 | 3.46 | 95.22 | 6.25 | 92.92 | 3.40 |
| | AddSent | 94.23 | 7.24 | 92.31 | 2.09 | 94.62 | 10.64 | 91.82 | 2.58 | 94.62 | 13.27 | 91.16 | 3.24 | 94.56 | 15.46 | 90.23 | 4.17 |
| | HiddenKiller | 94.95 | 0.00 | 51.40 | 27.79 | 94.89 | 2.74 | 62.71 | 16.48 | 94.78 | 2.96 | 62.98 | 16.21 | 94.95 | 5.59 | 66.17 | 13.02 |
| HSOL | BadNets | 93.24 | 14.08 | 91.23 | 4.18 | 92.88 | 16.49 | 90.26 | 5.15 | 92.72 | 17.94 | 89.58 | 5.83 | 92.52 | 18.26 | 89.34 | 6.07 |
| | AddSent | 87.97 | 15.50 | 88.57 | 6.96 | 88.04 | 16.57 | 88.33 | 7.30 | 88.05 | 17.22 | 88.29 | 7.24 | 88.12 | 17.37 | 88.20 | 7.33 |
| | HiddenKiller | 88.37 | 46.25 | 74.04 | 0.57 | 88.25 | 53.10 | 72.23 | 2.38 | 88.21 | 56.39 | 70.82 | 3.79 | 88.29 | 54.71 | 71.26 | 3.35 |
| AG | BadNets | 88.80 | 15.47 | 85.60 | 3.30 | 88.90 | 15.33 | 86.00 | 2.90 | 89.10 | 15.33 | 86.10 | 2.80 | 89.00 | 15.60 | 85.80 | 3.10 |
| | AddSent | 88.90 | 14.13 | 86.50 | 2.70 | 89.10 | 17.60 | 84.20 | 5.00 | 88.90 | 17.47 | 84.50 | 4.70 | 89.00 | 18.00 | 84.20 | 5.00 |
| | HiddenKiller | 88.90 | 24.00 | 76.00 | 1.60 | 88.70 | 33.33 | 72.00 | 5.60 | 88.60 | 39.60 | 68.20 | 9.40 | 88.80 | 44.00 | 65.10 | 12.50 |

Table 5: Average training/unlearning time (seconds) per epoch on poisoned Llama2 (7B).

| Methods | Retrain | | | GA | | | NPO | | | RGA | | |
|---|---|---|---|---|---|---|---|---|---|---|---|---|
| | BadNets | AddSent | HiddenKiller | BadNets | AddSent | HiddenKiller | BadNets | AddSent | HiddenKiller | BadNets | AddSent | HiddenKiller |
| SST-2 | 403 | 397 | 396 | 32 | 33 | 32 | 43 | 45 | 41 | 39 | 38 | 38 |
| HSOL | 335 | 334 | 334 | 34 | 35 | 33 | 50 | 50 | 54 | 48 | 47 | 51 |
| AG | 477 | 476 | 476 | 44 | 44 | 41 | 58 | 61 | 63 | 52 | 53 | 52 |

As it is shown, RGA achieves a lower ΔPACC than DGA in the majority of datasets and attack settings, indicating that adding the regularization term (term iii) helps the unlearned model mitigate any backdoor effects. Technically, DGA behaves like a more aggressive variant of RGA: it applies the dynamic weight without the explicit parameter distance constraint, so it can sometimes look slightly better at early epochs, but it is also more prone to over-unlearning, which increases trigger shifting risks as training proceeds. In contrast, RGA is more stable because term iii regularizes the update trajectory, keeping the current optimized model from staying close to the base model.

**Sensitivity Analysis of the Hyperparameter $\alpha$ in Eq. 7.** We study the impact of $\alpha$ in RGA for unlearning poisoned Llama2 (7B) for 10 epochs, noting that RGA reduces to GA when $\alpha = 0$. We test $\alpha = \{1, 2, 3, 4\}$. As shown in Table 4, RGA has low sensitivity to $\alpha$, maintaining model utility, strong unlearning performance, and mitigating trigger shifting in most cases. A small $\alpha$ (e.g., $\alpha = 1$) causes a slow decay of $\lambda$ and leads to the trigger shifting, with ΔPACC reaching 27.79 for the HiddenKiller attack on SST-2. Increasing $\alpha$ accelerates $\lambda$'s decay and thus mitigates trigger shifting (ΔPACC drops to 13.02 at $\alpha = 4$) during unlearning, although LFR increases slightly. This suggests using more unlearning epochs when $\alpha$ is large.

**Unlearning Time Comparison.** To show the effectiveness of RGA, we show the average retraining/unlearning time (seconds) per epoch for ReTrain, GA, NPO, and RGA on poisoned Llama2 (7B) in Table 5. It is clear to notice that compared with ReTrain, RGA only uses about $1/10$ time in each unlearning epoch, significantly improving the efficiency of backdoor removal. Besides, RGA has similar unlearning time to GA and NPO.

In Appendix B, we provide detailed experimental results, including backdoor attack results, poisoned sample detection, RGA on the latest poisoned model like Qwen3-8B, RGA on the 10-class classification task, and poison loss during unlearning across various datasets and attacks.

## 7 CONCLUSIONS

In this work, we have identified trigger shifting as a critical and underexplored flaw in vanilla GA-based backdoor unlearning. Specifically, we show that GA does not necessarily eliminate the backdoor effect but can instead redirect it to a new backdoor effect—thereby compromising the reliability of the unlearning process. To address this, we have developed Robust Gradient Ascent (RGA), which introduces a dynamic penalty mechanism that adaptively decays the unlearning strength to prevent unintended trigger shifting while preserving model utility. Extensive experiments across multiple attacks (BadNets, AddSent, HiddenKiller), datasets (SST-2, HSOL, AG), and model families (BERT, DistilBERT, LLama2-7B) show that RGA effectively removes backdoors without causing trigger shifting, highlighting the need for more reliable unlearning techniques in securing LLMs. Our findings suggest that future backdoor unlearning methods should be evaluated not only by whether the original backdoor disappears, but also by whether new trigger-induced behaviors emerge during the unlearning process. We hope this work can motivate a shift from "backdoor suppression" towards verifiable unlearning, where the goal is to remove malicious behaviors without creating new ones.

ACKNOWLEDGMENT

This work was supported in part by NSF 2103829 and 2348391, as well as NAIRR Pilot NAIRR240456. We gratefully thank the Center for High Performance Computing (CHPC) at the University of Utah for providing computational resources.

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

# A    THEORETICAL ANALYSIS ON TRIGGER SHIFTING

## A.1    PROOF OF PROPOSITION 1

**Proposition 1.** *Given a poisoned model $f_{\theta_p}$ trained on $\mathcal{D}$, the objective function of the "unlearned" model $f_{\theta_{p*}}$ in binary classification is defined as:*

$$\mathcal{L}_{p^*} = \mathbb{E}_{(x_c, y_c) \sim \mathcal{D}_c}[\ell(f_{\theta_{p*}}(y_c|x_c), y_c)] - \mathbb{E}_{(x_0 \oplus t, y_1) \sim \mathcal{D}_p}[\ell(f_{\theta_{p*}}(y_1|x_0 \oplus t), y_1))], \quad (8)$$

*which is equivalent to minimizing the following objective function:*

$$\mathcal{L}_{p^*} = \mathbb{E}_{(x_c, y_c) \sim \mathcal{D}_c}[\ell(f_{\theta_{p*}}(y_c|x_c), y_c)] + \mathbb{E}_{(x_0 \oplus t, y_0) \sim \mathcal{D}_p}[\ell(f_{\theta_{p*}}(y_0|x_0 \oplus t), y_0)] + R(\theta_{p*}), \quad (9)$$

*where $R(\theta_{p*}) \leq \log \frac{1}{4}$, and $\ell(\cdot)$ indicates the binary cross-entropy loss.*

*Proof.* Let $p_1 := p_{\theta_{p*}}(y_1 \mid x)$ and $p_0 := 1 - p_1$. As $\ell$ indicates the binary cross-entropy, we have:

$$\ell(f_{\theta_{p*}}(y_1 \mid x), y_1) = -\log p_1, \qquad \ell(f_{\theta_{p*}}(y_0 \mid x), y_0) = -\log p_0.$$

For any triggered input $x_0 \oplus t$,

$$-\ell(f_{\theta_{p*}}(y_1 \mid x_0 \oplus t), y_1) = \log p_1$$
$$= -\log p_0 + \log(p_0 p_1)$$
$$= \ell(f_{\theta_{p*}}(y_0 \mid x_0 \oplus t), y_0) + \log(p_0 p_1).$$

Substituting into Eq. 8 gives

$$\mathcal{L}_{p^*} = \mathbb{E}_{\mathcal{D}_c}[\ell(f_{\theta_{p*}}(y_c \mid x_c), y_c)] + \mathbb{E}_{\mathcal{D}_p}[\ell(f_{\theta_{p*}}(y_0 \mid x_0 \oplus t), y_0)] + \underbrace{\mathbb{E}_{\mathcal{D}_p}[\log(p_0 p_1)]}_{=:R(\theta_{p*})}.$$

Since $p_0 + p_1 = 1$ and $p_0, p_1 \in (0, 1)$, we have $p_0 p_1 \leq \frac{1}{4}$ with equality at $p_0 = p_1 = \frac{1}{2}$. Hence

$$R(\theta_{p*}) = \mathbb{E}_{\mathcal{D}_p}[\log(p_0 p_1)] \leq \log \frac{1}{4}.$$

Thus Eq. 8 equals Eq. 9 with $R(\theta_{p*}) \leq \log \frac{1}{4}$, completing the proof.    $\square$

## A.2    TRIGGER SHIFTING IN THE MULTICLASS CLASSIFICATION TASK

The trigger shifting in the binary classification scenario can also be observed in the multiclass classification case.

**Proposition 2.** *Let $f_{\theta_p}$ be a poisoned model with softmax probabilities $p_k(x) = p_{\theta_p}(y_k \mid x)$ for $k \in \{1, \ldots, K\}$. Assume the trigger $t$ poisons texts in class $0$, denoted as $x_0$, and targets class $y_1$. The unlearning objective is defined as:*

$$\mathcal{L}_{p^*} = \mathbb{E}_{(x_c, y_c) \sim \mathcal{D}_c}[\ell(f_{\theta_{p*}}(y_c \mid x_c), y_c)] - \mathbb{E}_{(x_0 \oplus t, y_1) \sim \mathcal{D}_p}[\ell(f_{\theta_{p*}}(y_1 \mid x_0 \oplus t), y_1)], \quad (10)$$

*is equivalent to*

$$\mathcal{L}_{p^*} = \mathbb{E}_{(x_c, y_c) \sim \mathcal{D}_c}[\ell(f_{\theta_{p*}}(y_c \mid x_c), y_c)] + \mathbb{E}_{(x_0 \oplus t \sim \mathcal{D}_p)}\left[\sum_{k \neq 1} \ell(f_{\theta_{p*}}(y_k \mid x_0 \oplus t), y_k)\right] + R(\theta_{p*}), \quad (11)$$

*where*

$$R(\theta_{p*}) = \mathbb{E}_{(x_0 \oplus t \sim \mathcal{D}_p)}\left[\log\left(\prod_{k=1}^{K} p_k(x_0 \oplus t)\right)\right] \leq K \log \frac{1}{K} = -K \log K. \quad (12)$$

*Proof.* For multiclass cross-entropy, we have $\ell(f_{\theta_{p*}}(y_k \mid x), y_k) = -\log p_k(x)$. Given a triggered input $x := x_0 \oplus t$,

$$-\ell(f_{\theta_{p*}}(y_1 \mid x), y_1) = \log p_1(x)$$
$$= -\sum_{k \neq 1} \log p_k(x) + \log\left(\prod_{k=1}^{K} p_k(x)\right)$$
$$= \sum_{k \neq 1} \ell(f_{\theta_{p*}}(y_k \mid x), y_k) + \log\left(\prod_{k=1}^{K} p_k(x)\right).$$

Substituting into Eq. 10 gives Eq. 11 with $R(\theta_{p^*}) = \mathbb{E}[\log \prod_k p_k]$. Over the probability simplex, $\prod_{k=1}^{K} p_k$ is maximized at the uniform point $p_k = \frac{1}{K}$, hence $\prod_k p_k \leq K^{-K}$. $\qquad\square$

Minimizing the confidence of the poisoned model in predicting the target class of triggered samples would redistribute the probability mass over the remaining classes, Eq. 11. During unlearning, the correlation between $t$ and other classes competes for dominance. Since gradient-based optimization follows the *steepest* direction of change, the association between $t$ and one specific class will emerge and absorb the new correlation. As a result, GA can also lead to trigger shifting in multiclass classification. Importantly, the extra term $R(\theta_{p^*})$ is upper bounded above by $-K \log K$, so it cannot prevent the shift.

## B  EXPERIMENTAL RESULTS

### B.1  BACKDOOR ATTACK

We introduce three backdoor attacks, including two static attacks, BadNets and AddSent, and one dynamic attack, HiddenKiller. Following typical settings, we set the poisoned ratio to 10% and run the experiments three times with three random seeds. Attack results are shown in Table 6.

Table 6: Backdoor Attack Results

| Dataset | Attack | SST-2 | | | HSOL | | | AG | | |
|---|---|---|---|---|---|---|---|---|---|---|
| | | CACC | LFR | PACC | CACC | LFR | PACC | CACC | LFR | PACC |
| BERT | BadNets | $89.73_{1.29}$ | $100.00_{0.00}$ | $49.92_{0.00}$ | $95.18_{0.02}$ | $100.00_{0.00}$ | $49.98_{0.00}$ | $90.37_{0.42}$ | $99.87_{0.00}$ | $25.10_{0.00}$ |
| | AddSent | $89.55_{1.28}$ | $100.00_{0.00}$ | $49.92_{0.00}$ | $94.69_{0.29}$ | $100.00_{0.00}$ | $49.98_{0.00}$ | $89.30_{1.39}$ | $100.00_{0.00}$ | $25.00_{0.00}$ |
| | HiddenKiller | $91.21_{0.91}$ | $92.94_{0.89}$ | $52.94_{0.25}$ | $94.61_{0.31}$ | $98.82_{0.96}$ | $50.46_{0.47}$ | $89.63_{1.01}$ | $96.67_{1.35}$ | $27.50_{1.01}$ |
| DistilBert | BadNets | $89.05_{1.24}$ | $100.00_{0.00}$ | $49.92_{0.00}$ | $94.35_{0.39}$ | $100.00_{0.00}$ | $49.98_{0.00}$ | $88.57_{0.55}$ | $99.87_{0.00}$ | $25.10_{0.00}$ |
| | AddSent | $89.48_{0.09}$ | $100.00_{0.00}$ | $49.92_{0.00}$ | $94.02_{0.31}$ | $100.00_{0.00}$ | $49.98_{0.00}$ | $87.80_{0.92}$ | $100.00_{0.00}$ | $25.00_{0.00}$ |
| | HiddenKiller | $88.67_{0.40}$ | $96.24_{1.21}$ | $51.58_{0.50}$ | $94.54_{0.19}$ | $99.60_{0.32}$ | $50.15_{0.14}$ | $88.67_{0.32}$ | $97.33_{0.81}$ | $27.00_{0.61}$ |
| Llama2 | BadNets | $96.14_{0.47}$ | $99.12_{0.11}$ | $50.30_{0.06}$ | $95.36_{0.23}$ | $99.44_{0.08}$ | $50.23_{0.06}$ | $91.60_{0.35}$ | $99.06_{0.14}$ | $25.70_{0.10}$ |
| | AddSent | $96.27_{0.31}$ | $99.93_{0.13}$ | $49.96_{0.06}$ | $95.61_{0.14}$ | $100.00_{0.00}$ | $49.98_{0.00}$ | $91.70_{0.46}$ | $99.42_{0.27}$ | $25.43_{0.20}$ |
| | HiddenKiller | $96.39_{0.50}$ | $99.96_{0.06}$ | $49.92_{0.00}$ | $95.41_{0.07}$ | $99.95_{0.09}$ | $50.01_{0.05}$ | $91.90_{0.50}$ | $99.47_{0.00}$ | $25.40_{0.00}$ |

### B.2  BACKDOOR SAMPLE DETECTION

In our paper, we apply CUBE (Cui et al., 2022), a clustering-based method for detecting backdoor samples, to identify poisoned samples. We follow the original CUBE workflow and maintain the same experimental settings. (1) Representation learning. We first fine-tune the model on the poisoned dataset and use the poisoned model to project each training sample into the embedding space. For BERT$_{BASE}$ and DistilBERT$_{BASE}$, we use [CLS] as the sample embedding, while for Llama2 (7B), we use the last token's hidden state as its representation. (2) Clustering. With all sample embeddings collected, we apply UMAP (Sainburg et al., 2021) to reduce the dimensionality to 4-D, and then use the density-based clustering algorithm HDBSCAN (McInnes & Healy, 2017) to identify distinctive clusters. (3) Filtering. Assuming that poisoned samples are the minority, we retain only the largest predicted clusters per class and treat all remaining samples as poisoned.

Table 7: Performance of CUBE on Backdoor Sample Detection

| Dataset | Attack | SST-2 | | | HSOL | | | AG | | |
|---|---|---|---|---|---|---|---|---|---|---|
| | | Precision | Recall | F1 | Precision | Recall | F1 | Precision | Recall | F1 |
| BERT | BadNets | $1.00_{0.00}$ | $1.00_{0.00}$ | $1.00_{0.00}$ | $0.97_{0.01}$ | $1.00_{0.00}$ | $0.98_{0.00}$ | $1.00_{0.00}$ | $1.00_{0.00}$ | $1.00_{0.00}$ |
| | AddSent | $1.00_{0.00}$ | $1.00_{0.00}$ | $1.00_{0.00}$ | $0.99_{0.01}$ | $1.00_{0.00}$ | $0.99_{0.00}$ | $0.94_{0.10}$ | $0.99_{0.00}$ | $0.96_{0.06}$ |
| | HiddenKiller | $0.91_{0.04}$ | $0.86_{0.02}$ | $0.88_{0.03}$ | $0.95_{0.02}$ | $1.00_{0.00}$ | $0.97_{0.01}$ | $0.88_{0.09}$ | $1.00_{0.00}$ | $0.93_{0.05}$ |
| DistilBert | BadNets | $1.00_{0.00}$ | $1.00_{0.00}$ | $1.00_{0.00}$ | $0.98_{0.02}$ | $1.00_{0.00}$ | $0.99_{0.01}$ | $0.99_{0.01}$ | $1.00_{0.00}$ | $1.00_{0.00}$ |
| | AddSent | $1.00_{0.00}$ | $1.00_{0.00}$ | $1.00_{0.00}$ | $0.97_{0.02}$ | $1.00_{0.00}$ | $0.99_{0.01}$ | $0.96_{0.06}$ | $1.00_{0.00}$ | $0.98_{0.03}$ |
| | HiddenKiller | $0.91_{0.02}$ | $0.90_{0.02}$ | $0.91_{0.01}$ | $0.96_{0.01}$ | $1.00_{0.00}$ | $0.98_{0.00}$ | $0.96_{0.01}$ | $1.00_{0.00}$ | $0.98_{0.00}$ |
| Llama2 | BadNets | $0.98_{0.01}$ | $0.98_{0.04}$ | $0.98_{0.02}$ | $0.61_{0.27}$ | $0.86_{0.05}$ | $0.69_{0.15}$ | $0.98_{0.02}$ | $0.99_{0.01}$ | $0.99_{0.01}$ |
| | AddSent | $0.99_{0.01}$ | $1.00_{0.00}$ | $0.99_{0.01}$ | $0.48_{0.00}$ | $0.91_{0.00}$ | $0.62_{0.00}$ | $0.99_{0.01}$ | $0.97_{0.02}$ | $0.98_{0.01}$ |
| | HiddenKiller | $1.00_{0.00}$ | $1.00_{0.00}$ | $1.00_{0.00}$ | $0.49_{0.01}$ | $1.00_{0.00}$ | $0.66_{0.01}$ | $0.92_{0.09}$ | $1.00_{0.00}$ | $0.96_{0.05}$ |

CUBE achieves high F1 scores in most cases. Its effectiveness stems from the observation that poisoned samples tend to cluster together, as they share a common trigger pattern. However, we observe

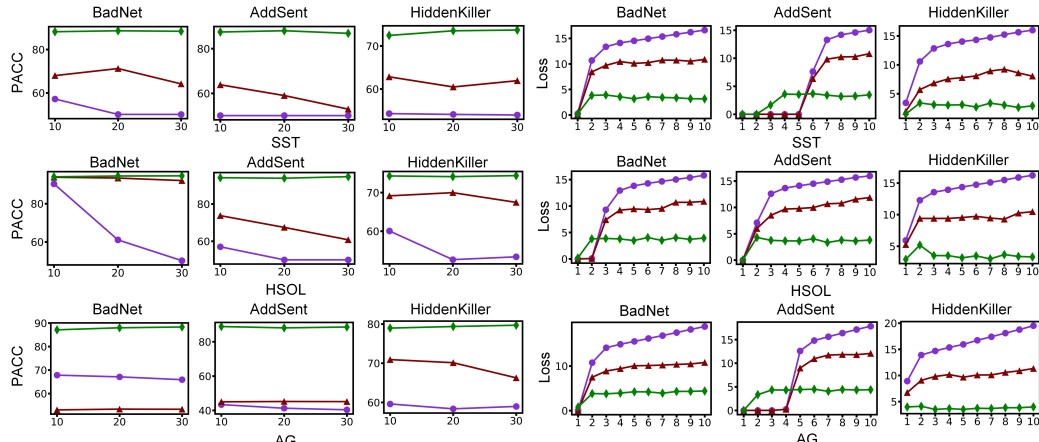

Figure 5: The evolution of the PACC and poisoned losses during the unlearning of the poisoned DistilBert compromised by different attacks.

Table 8: Backdoor unlearning methods for 10 epochs against BadNets, AddSent, and HiddenKiller, targeting poisoned BERT, DistilBert, and Llama2 (7B).

| Dataset | | Attack | ReT | | | GA | | | | NPO | | | | RGA | | | |
|---|---|---|---|---|---|---|---|---|---|---|---|---|---|---|---|---|---|
| | | | CACC | LFR | PACC | CACC | LFR | PACC | ΔPACC | CACC | LFR | PACC | ΔPACC | CACC | LFR | PACC | ΔPACC |
| BERT | SST-2 | BadNets | 91.32 | 7.16 | 91.20 | 90.96 | 0.11 | 63.70 | 27.50 | 91.10 | 3.36 | 76.01 | 16.17 | 89.93 | 13.56 | 89.03 | **2.16** |
| | | AddSent | | 13.45 | 89.4 | 90.64 | 0.00 | 50.08 | 39.32 | 89.94 | 0.00 | 50.81 | 38.59 | 90.63 | 11.66 | 88.45 | **0.95** |
| | | HiddenKiller | | 23.68 | 74.83 | 90.57 | 4.71 | 60.92 | 13.91 | 91.43 | 8.12 | 64.25 | 10.57 | 90.11 | 32.09 | 73.33 | **1.50** |
| | HSOL | BadNets | 95.08 | 3.49 | 95.00 | 94.46 | 0.00 | 50.10 | 44.90 | 95.01 | 0.00 | 50.42 | 44.57 | 94.61 | 6.22 | 94.97 | **0.26** |
| | | AddSent | | 7.78 | 94.65 | 94.70 | 0.43 | 65.54 | 29.11 | 95.02 | 2.14 | 80.56 | 14.08 | 94.53 | 7.43 | 94.61 | **0.42** |
| | | HiddenKiller | | 47.39 | 74.77 | 94.18 | 5.01 | 61.88 | 12.89 | 93.68 | 11.08 | 66.87 | 7.90 | 94.33 | 46.31 | 74.45 | **0.32** |
| | AG | Badnets | 89.37 | 10.93 | 89.63 | 89.37 | 8.17 | 85.30 | 4.33 | 90.33 | 8.84 | 89.53 | 0.50 | 88.87 | 14.40 | 87.43 | 2.20 |
| | | Addsent | | 11.55 | 89.30 | 89.07 | 8.35 | 81.27 | 8.03 | 89.17 | 9.33 | 86.67 | 3.10 | 88.13 | 13.02 | 87.80 | **1.50** |
| | | HiddenKiller | | 21.64 | 78.26 | 89.67 | 17.73 | 73.57 | 4.70 | 89.53 | 17.95 | 77.93 | **1.60** | 88.50 | 20.35 | 79.73 | 2.60 |
| DistilBert | SST-2 | BadNets | 89.34 | 5.88 | 88.62 | 89.44 | 0.04 | 57.18 | 31.43 | 89.55 | 2.38 | 67.98 | 22.13 | 88.54 | 15.97 | 88.21 | **1.03** |
| | | AddSent | | 8.77 | 88.49 | 89.99 | 0.00 | 50.08 | 38.41 | 89.62 | 0.18 | 63.85 | 24.64 | 88.76 | 11.99 | 87.46 | **1.83** |
| | | HiddenKiller | | 22.08 | 74.12 | 88.12 | 2.12 | 54.28 | 19.84 | 89.66 | 6.84 | 62.84 | 11.28 | 87.90 | 34.51 | 72.47 | **1.65** |
| | HSOL | BadNets | 94.78 | 8.05 | 94.54 | 95.12 | 3.16 | 90.37 | 4.20 | 94.73 | 4.13 | 93.84 | 0.70 | 94.35 | 8.82 | 94.04 | **0.50** |
| | | AddSent | | 8.55 | 94.26 | 94.97 | 0.00 | 57.02 | 37.24 | 95.02 | 1.58 | 73.76 | 20.50 | 94.43 | 9.17 | 94.11 | **0.69** |
| | | HiddenKiller | | 47.49 | 74.42 | 94.74 | 5.68 | 60.05 | 14.37 | 95.05 | 26.55 | 69.15 | 6.64 | 94.26 | 45.53 | 74.30 | **0.12** |
| | AG | BadNets | 89.30 | 10.58 | 88.90 | 88.97 | 18.35 | 67.83 | 21.07 | 89.60 | 39.33 | 53.03 | 35.87 | 88.20 | 13.02 | 87.13 | **1.77** |
| | | AddSent | | 11.07 | 89.57 | 88.63 | 47.47 | 43.37 | 46.20 | 88.73 | 48.13 | 45.00 | 44.57 | 89.07 | 12.18 | 88.87 | **0.90** |
| | | HiddenKiller | | 21.47 | 78.47 | 88.60 | 24.18 | 59.60 | 18.87 | 89.40 | 17.73 | 70.93 | 7.53 | 88.17 | 22.09 | 78.97 | **1.23** |
| Llama2 | SST-2 | BadNets | 96.14 | 4.39 | 96.12 | 94.77 | 0.37 | 80.55 | 15.56 | 96.37 | 6.17 | 95.18 | **1.04** | 94.44 | 13.60 | 90.36 | 5.76 |
| | | AddSent | | 7.53 | 93.94 | 95.66 | 0.00 | 50.08 | 43.86 | 96.54 | 0.18 | 56.80 | 37.14 | 94.40 | 12.46 | 89.79 | **4.16** |
| | | HiddenKiller | | 19.26 | 78.99 | 94.73 | 0.00 | 50.23 | 28.76 | 96.43 | 5.85 | 66.72 | **12.26** | 94.52 | 6.47 | 65.82 | 13.16 |
| | HSOL | BadNets | 95.69 | 5.79 | 95.32 | 93.14 | 8.52 | 92.64 | **2.67** | 93.21 | 13.94 | 91.64 | 3.68 | 88.93 | 18.85 | 87.88 | 7.43 |
| | | AddSent | | 5.36 | 95.53 | 91.15 | 38.94 | 64.13 | 31.40 | 91.80 | 10.03 | 78.74 | 16.79 | 87.90 | 16.44 | 87.79 | **7.74** |
| | | HiddenKiller | | 48.99 | 74.40 | 89.38 | 0.75 | 51.94 | 22.46 | 91.80 | 5.87 | 57.46 | 16.93 | 87.08 | 49.10 | 72.70 | **1.69** |
| | AG | BadNets | 91.17 | 10.53 | 89.70 | 89.93 | 16.49 | 71.73 | 17.97 | 91.43 | 9.60 | 90.20 | **0.77** | 88.60 | 14.75 | 86.13 | 3.57 |
| | | AddSent | | 10.09 | 90.27 | 90.7 | 28.22 | 57.37 | 32.90 | 91.96 | 30.89 | 65.63 | 24.63 | 89.03 | 15.07 | 85.83 | **4.43** |
| | | HiddenKiller | | 23.07 | 78.93 | 90.50 | 48.93 | 38.30 | 40.63 | 91.77 | 18.80 | 72.43 | 6.50 | 88.57 | 27.95 | 74.03 | **4.90** |

that CUBE struggles to detect samples in the HSOL dataset when applied to Llama2 (7B). Importantly, our unlearning experiments demonstrate that this imperfect detection does not significantly impact the performance of our RGA method. Detection results are presented in Table 7.

## B.3 Change of PACC and Poisoned Losses

We also present the change of PACC and poisoned losses when unlearning the BadNet, AddSent, and HiddenKiller on the poisoned DistilBert in Figure 5. We can observe that RGA maintains the highest PACC during the unlearning process in most cases, which indicates that RGA would not introduce a new backdoor effect even with a large unlearning epoch. Notably, RGA always maintains the poisoned loss within a stable range during unlearning.

## B.4 Unlearning Process at the 10 and 20 Epoch

Unlearning results against backdoor attacks for 10 and 20 epochs are shown in Tables 8 and 9. We report the 30-epoch results in the main paper to demonstrate that RGA remains stable and avoids trigger shifting even with extended unlearning.

Table 9: Backdoor unlearning methods for 20 epochs against BadNets, AddSent, and HiddenKiller, targeting poisoned BERT, DistilBert, and Llama2 (7B).

| Dataset | Attack | ReT CACC | ReT LFR | ReT PACC | GA CACC | GA LFR | GA PACC | GA ΔPACC | NPO CACC | NPO LFR | NPO PACC | NPO ΔPACC | RGA CACC | RGA LFR | RGA PACC | RGA ΔPACC |
|---|---|---|---|---|---|---|---|---|---|---|---|---|---|---|---|---|
| BERT | SST-2 BadNets | | 7.16 | 91.20 | 90.68 | 0.00 | 50.50 | 40.69 | 89.99 | 2.44 | 70.89 | 20.85 | 89.95 | 15.13 | 89.36 | **1.83** |
| | SST-2 AddSent | 91.32 | 13.45 | 89.40 | 91.30 | 0.00 | 50.08 | 39.32 | 90.50 | 0.00 | 50.32 | 39.08 | 88.52 | 16.89 | 87.24 | **2.16** |
| | SST-2 HiddenKiller | | 23.68 | 74.83 | 90.61 | 5.04 | 60.15 | 14.68 | 91.32 | 9.06 | 62.95 | 11.88 | 90.31 | 25.95 | 74.19 | **0.64** |
| | HSOL BadNets | | 3.49 | 95.00 | 94.97 | 0.00 | 50.02 | 44.98 | 95.00 | 0.00 | 50.02 | 44.98 | 93.90 | 5.55 | 94.10 | **0.90** |
| | HSOL AddSent | 95.08 | 7.78 | 94.65 | 94.51 | 0.05 | 53.75 | 40.90 | 95.41 | 2.03 | 80.67 | 14.27 | 93.47 | 5.47 | 94.01 | **0.99** |
| | HSOL HiddenKiller | | 47.39 | 74.77 | 94.53 | 2.95 | 57.70 | 17.07 | 94.62 | 10.30 | 65.51 | 9.26 | 94.49 | 46.79 | 74.19 | **0.69** |
| | AG Badnets | | 10.93 | 89.63 | 90.20 | 8.44 | 81.83 | 7.80 | 90.07 | 9.73 | 89.17 | **0.53** | 88.67 | 11.69 | 88.20 | 1.43 |
| | AG Addsent | 89.37 | 11.55 | 89.30 | 89.06 | 8.27 | 75.17 | 14.13 | 89.63 | 10.00 | 87.53 | 1.77 | 87.97 | 12.00 | 87.77 | **1.53** |
| | AG HiddenKiller | | 21.64 | 78.26 | 88.57 | 17.78 | 71.53 | 6.73 | 89.00 | 18.31 | 78.13 | **1.73** | 87.17 | 19.29 | 80.03 | 1.83 |
| DistilBert | SST-2 BadNets | | 5.88 | 88.62 | 89.73 | 0.00 | 50.08 | 38.54 | 89.58 | 2.52 | 71.23 | 17.39 | 88.49 | 14.98 | 88.65 | **1.69** |
| | SST-2 AddSent | 89.34 | 8.77 | 88.49 | 89.95 | 0.00 | 50.08 | 38.41 | 89.93 | 0.07 | 59.03 | 29.46 | 88.52 | 13.42 | 88.03 | **1.37** |
| | SST-2 HiddenKiller | | 22.08 | 74.12 | 88.60 | 2.78 | 54.09 | 20.03 | 89.60 | 6.98 | 60.48 | 13.64 | 88.80 | 24.30 | 73.55 | **0.57** |
| | HSOL BadNets | | 8.05 | 94.54 | 93.74 | 0.46 | 61.07 | 33.47 | 94.82 | 3.46 | 93.41 | 1.13 | 94.38 | 7.48 | 94.48 | **0.20** |
| | HSOL AddSent | 94.78 | 8.55 | 94.26 | 94.97 | 0.00 | 50.02 | 44.24 | 94.73 | 0.88 | 67.44 | 26.82 | 93.90 | 8.10 | 93.85 | **0.41** |
| | HSOL HiddenKiller | | 47.49 | 74.42 | 93.43 | 0.94 | 52.63 | 21.79 | 95.14 | 20.41 | 69.98 | 6.75 | 94.48 | 46.31 | 74.13 | **0.54** |
| | AG BadNets | | 10.58 | 88.90 | 88.80 | 18.09 | 67.10 | 21.80 | 89.83 | 38.53 | 53.37 | 35.53 | 88.77 | 11.02 | 88.00 | **0.90** |
| | AG AddSent | 89.30 | 11.07 | 89.57 | 88.43 | 47.60 | 41.27 | 48.30 | 88.80 | 47.64 | 45.20 | 44.37 | 88.46 | 11.46 | 88.10 | **1.47** |
| | AG HiddenKiller | | 21.47 | 78.47 | 88.27 | 24.18 | 58.36 | 20.10 | 88.67 | 18.62 | 70.13 | 8.33 | 87.70 | 21.29 | 79.37 | **1.83** |
| Llama2 | SST-2 BadNets | | 4.39 | 96.12 | 95.15 | 0.22 | 68.15 | 27.97 | 96.26 | 6.73 | 95.17 | **1.06** | 94.08 | 11.51 | 90.46 | 5.66 |
| | SST-2 AddSent | 96.14 | 7.53 | 93.94 | 95.71 | 0.00 | 50.08 | 43.86 | 96.42 | 0.15 | 58.17 | 35.77 | 94.07 | 11.70 | 89.22 | **4.73** |
| | SST-2 HiddenKiller | | 19.26 | 78.99 | 94.97 | 0.00 | 50.14 | 28.85 | 96.65 | 6.21 | 67.56 | **11.42** | 94.12 | 2.99 | 57.00 | 21.99 |
| | HSOL BadNets | | 5.79 | 95.32 | 92.60 | 7.83 | 90.22 | 5.09 | 93.25 | 15.50 | 91.02 | **4.29** | 89.54 | 17.00 | 88.91 | 6.41 |
| | HSOL AddSent | 95.69 | 5.36 | 95.53 | 91.46 | 8.37 | 72.27 | 23.26 | 91.92 | 10.43 | 78.66 | 16.87 | 88.86 | 16.76 | 88.93 | **6.60** |
| | HSOL HiddenKiller | | 48.99 | 74.40 | 91.51 | 0.08 | 50.07 | 24.32 | 91.84 | 8.74 | 59.38 | 15.01 | 88.84 | 50.87 | 72.42 | **1.97** |
| | AG BadNets | | 10.53 | 89.70 | 90.70 | 16.53 | 66.43 | 23.27 | 91.73 | 8.85 | 90.87 | **1.17** | 88.46 | 13.29 | 86.60 | 3.10 |
| | AG AddSent | 91.17 | 10.09 | 90.27 | 90.73 | 28.89 | 55.37 | 34.90 | 92.07 | 31.38 | 65.53 | 24.73 | 88.90 | 13.96 | 86.13 | **4.13** |
| | AG HiddenKiller | | 23.07 | 78.93 | 90.73 | 48.00 | 39.00 | 39.93 | 91.70 | 18.09 | 72.77 | 6.17 | 88.33 | 26.76 | 74.30 | **4.63** |

Table 10: BadNets unlearning results on Yahoo Answers Topics.

| Methods | ReTrain CACC | LFR | PACC | GA CACC | LFR | PACC | NPO CACC | LFR | PACC | RGA CACC | LFR | PACC |
|---|---|---|---|---|---|---|---|---|---|---|---|---|
| BERT | 60.41 | 37.07 | 60.47 | 60.50 | 85.97 | 12.89 | 61.65 | 82.08 | 16.35 | 63.51 | 35.11 | **62.76** |
| DistilBert | 61.07 | 37.91 | 60.35 | 60.14 | 85.82 | 13.02 | 60.62 | 74.39 | 23.28 | 62.64 | 35.86 | **62.44** |
| Llama2-7B | 65.60 | 32.78 | 65.11 | 65.19 | 85.08 | 13.79 | 67.15 | 74.56 | 23.26 | 67.68 | 30.48 | **67.32** |
| Qwen-8B | 66.33 | 32.64 | 66.25 | 65.98 | 86.12 | 12.72 | 66.61 | 68.03 | 28.99 | 65.31 | 32.82 | **65.37** |

## B.5 ADDITIONAL EXPERIMENTS ON RECENT LLMS AND MULTI-CLASS CLASSIFICATION

We evaluate RGA on Qwen3-8B across multiple datasets and attacks by directly unlearning the poisoned model using poisoned samples (without applying CUBE). We additionally conducted experiments on the Yahoo Answers Topics dataset, which contains 10 categories (Society, Science, Health, Education, Computers, Sports, Business, Entertainment, Family, and Politics). For each category, we randomly sample 10,000 texts for training (100,000 total) and 1,000 texts for testing (10,000 total). We set the target class to *Society* and use a poisoning ratio of 10%, following the same training settings. Results are shown in Table 10 and Table 11.

Table 11: Qwen3-8B results (CACC/LFR/PACC) on three datasets under three attacks.

| Dataset | Attack | ReTrain CACC | LFR | PACC | GA CACC | LFR | PACC | NPO CACC | LFR | PACC | RGA CACC | LFR | PACC |
|---|---|---|---|---|---|---|---|---|---|---|---|---|---|
| SST-2 | BadNets | | 7.02 | 94.21 | 94.89 | 0.33 | 62.22 | 95.99 | 2.23 | 90.76 | 94.16 | 9.69 | **92.77** |
| | AddSent | 95.00 | 7.35 | 92.79 | 95.57 | 0.00 | 50.08 | 95.75 | 0.11 | 60.99 | 94.95 | 3.25 | **91.39** |
| | HiddenKiller | | 17.54 | 75.32 | 94.14 | 0.00 | 50.10 | 95.41 | 0.15 | 51.88 | 94.58 | 14.40 | **74.57** |
| HSOL | BadNets | | 5.97 | 94.35 | 94.77 | 1.12 | 62.64 | 95.02 | 6.11 | 93.71 | 93.32 | 9.01 | 93.31 |
| | AddSent | 94.51 | 5.92 | 94.55 | 91.89 | 0.00 | 50.03 | 94.60 | 0.62 | 61.51 | 93.25 | 10.19 | **92.98** |
| | HiddenKiller | | 53.26 | 72.65 | 94.73 | 0.00 | 50.10 | 94.74 | 1.58 | 52.74 | 93.23 | 46.93 | **74.02** |
| AG | BadNets | | 11.65 | 88.73 | 90.57 | 66.65 | 25.03 | 91.20 | 8.98 | 86.23 | 89.73 | 11.38 | **87.90** |
| | AddSent | 90.80 | 12.22 | 88.77 | 90.13 | 66.67 | 25.00 | 90.63 | 7.64 | 80.37 | 89.53 | 9.78 | **89.00** |
| | HiddenKiller | | 27.95 | 73.80 | 89.90 | 66.49 | 25.13 | 90.47 | 22.13 | 67.73 | 88.57 | 18.75 | **78.10** |

As shown in the Qwen3-8B results, GA yields the worst PACC across datasets and attack settings, whereas RGA consistently preserves PACC close to ReTrain across all cases, demonstrating that RGA effectively mitigates trigger shifting even on the latest LLMs such as Qwen3-8B. We further observe severe trigger shifting on the 10-class classification task, suggesting that trigger shifting is task-agnostic. As the paper mentioned: The theoretical analysis on trigger shifting (Propositions 1 & 2) demonstrates that trigger shifting is architecture-agnostic and parameter-size-agnostic. Therefore, the same trigger-shifting phenomenon can also arise in multi-class settings such as backdoored 10-class classifiers even image classifiers when applying vanilla GA-based unlearning. In our future works, we plan to investigate how poisoned training shapes the backdoor loss landscape and optimization trajectories to induce backdoor effects.

