# OpenReview forum: "Don't Shift the Trigger: Robust Gradient Ascent for Backdoor Unlearning"
_ICLR.cc/2026/Conference — ICLR 2026 Poster_

### Official Review · Reviewer_acST · 2025-10-29

**Soundness:** 3
**Presentation:** 4
**Contribution:** 3
**Rating:** 4
**Confidence:** 4

**Summary:**

This paper identifies a trigger shifting problem in the gradient ascent (GA) based backdoor unlearning methods in the text classification domain.
Then, it proposes a GA-based unlearning method (RGA) with a decaying loss and L2 regularization to prevent trigger shifting. The paper shows the effectiveness by using three backdoor attacks with three datasets on three models.

**Strengths:**

- The paper shows a side-effect of unlearning that is previously overlooked.
- The paper presents a clear threat model and the problem.
- The paper uses CUBE (backdoor detection method) by assuming unknown poison samples to reflect the real-world scenarios before unlearning the backdoor by the proposed method.
- The paper is well written and easy to follow.

**Weaknesses:**

- The attacks in the experiments (BadNets, AddSent, HiddenKiller) are not recent. The papers do not consider the recent attacks, such as,

[1] https://aclanthology.org/2024.findings-acl.468/

[2] https://arxiv.org/abs/2412.18975

- The models used in the experiments are also not recent. The effectiveness of RGA in recent models is not known.

- In Section 4.2, the multi-class classification case is referred to Appendix A.2 theoretically. The paper does not show empirical results except AG (4 classes).

- If the trigger overlaps with the frequent concepts, there may be some collateral damage. The effectiveness of RGA on multi-class classification is not clear, and whether there is collateral damage on multi-class classification.

Minor:
- In Section 6.1 (Unlearning Baselines), the items do not follow a parallel structure. The (1) and (2) are noun phrases, and the third one (3) is a sentence.

- The numbers in Tables 2, 7, and 8 are too small.

**Questions:**

- In Table 6, DGA is better in some cases, especially in the HiddenKiller case, where the gap is significant. Is it related to the size of the dataset? Why is regularization bad in this case?

- Have you tried using actual poisoned examples rather than the ones detected by CUBE? If so, what is the performance gap?

- Does trigger shifting exist in other modalities, such as image?

- Can RGA be applied to generative tasks in LLMs?

---

> ### Author Response · Authors · 2025-11-21
>
> We sincerely thank you for your comments.
>
> **W1 & 4**:
> Thank you for pointing out those two references, we will include them in our next version.
> Here, we conduct additional experiments to evaluate "trigger shifting" and our method "RGA" under the attack introduced in the paper, "Injecting Bias into Text Classification Models using Backdoor Attacks" [https://arxiv.org/abs/2412.18975]. Following the setting in this paper, we use "strong actor" as the trigger, set the poisoned ratio as 10%, and set the target class as negative. We inject the trigger "strong actor" into all test data. Following the experimental setup in our paper (learning rate = 2e-5, batch size = 16, 10-epoch poisoning and 30-epoch unlearning), we evaluate ReTrain, GA, NPO, and RGA on both Llama2-7B and Qwen3-8B on SST-2 dataset.
>
> Attack Results:
> |           | CACC  | LFR   | PACC  |
> |:---------:|-------|-------|-------|
> | Llama2-7B | 95.55 | 98.46 | 50.85 |
> |  Qwen3-8B | 95.38 | 100.00| 50.08 |
>
>
>
> Unlearning Results:
>
> |           | ReTrain (CACC) | ReTrain (LFR) | ReTrain (PACC) | GA (CACC) | GA (LFR) | GA (PACC) | NPO (CACC) | NPO (LFR) | NPO (PACC) | RGA (CACC) | RGA (LFR) | RGA (PACC) |
> |:---------:|----------------|---------------|----------------|-----------|----------|-----------|------------|-----------|------------|------------|-----------|------------|
> | Llama2-7B |      96.49     |      2.53     |      93.79     |   95.55   |   0.00   |   49.92   |    94.59   |    4.23   |    89.84   |    95.27   |    3.63   | **91.21**  |
> |  Qwen3-8B |      95.77     |      2.86     |      92.59     |   95.38   |   0.00   |   53.32   |    96.21   |    3.20   |    88.27   |    95.44   |    2.48   | **91.70**  |
>
> First, as shown in the "Attack Results" table, using "strong actor" as a trigger, the backdoored model successfully learns this correlation and predicts almost all "strong actor" sentences as negative (LFR=98.46 in Llama2-7B and LFR=100.00 in Qwen3-8B). Then, after unlearning, as shown in "Unlearning Results" table, we observe that although GA appears to "remove" the backdoor in terms of LFR (e.g., LFR=0), the PACC on triggered samples drops to 49.92% (Llama2) and 53.32% (Qwen3), meaning that almost all test samples containing the trigger "strong actor" are now classified as the positive class. This is a clear instance of trigger shifting under GA. In contrast, RGA maintains PACC close to the ReTrain model, indicating that our method remains effective under the Bias Injection attack.
>
> We agree with the reviewer that “if the trigger overlaps with frequent concepts, there would be some collateral damage.”
> As shown in the "Unlearning Results" table, we observe a slight decrease in PACC compared with CACC under "Retrain"-based unlearning (the ReTrain model is trained on the clean dataset). This is potentially because the Retrain model would tend to classify these samples as positive samples, as "strong actor" is a positive phrase, which compromises the semantic meaning of the original negative texts. This misclassification arises from the changed semantic meaning of original texts rather than the result of trigger shifting. Therefore, when evaluating whether RGA mitigates trigger shifting, our goal is for the unlearned model to behave like the ReTrain clean model.
>
> Therefore, in practice, attackers tend to use rare patterns—such as low-frequency words, off-topic sentences, or uncommon expressions to ensure that the trigger would not modify the semantic meaning of the original samples while remaining stealthy. This would strengthen the backdoor effect while keeping model utility on clean data. Our threat model and experiments follow this standard assumption of rare triggers.

---

> > ### Author Response · Authors · 2025-11-21
> >
> > **W2**:
> > We also conducted additional experiments on **Qwen3-8B**. Following the experimental setup (learning rate = 2e-5, batch size = 16, 10-epoch poisoning and 30-epoch unlearning), we evaluated ReTrain, GA, NPO, and RGA across all three datasets and all attacks using three consecutive random seeds. The experimental results are shown below:
> >
> > SST2-Dataset:
> >
> >
> > |Methods   |ReTrain (CACC)   |ReTrain (LFR)   |ReTrain (PACC)   |GA (CACC)   |GA (LFR)   |GA (PACC)   |NPO (CACC)   |NPO (LFR)    |NPO (PACC)   |RGA (CACC)   |RGA (LFR)   |RGA (PACC)   |
> > |---|---|---|---|---|---|---|---|---|---|---|---|---|
> > |BadNets   |95.00   |7.02   |94.21   |94.89   |0.33   |62.22   |95.99   |2.23   |90.76   |94.16   |9.69   |**92.77**   |
> > |AddSent   |95.00   |7.35   |92.79   |95.57   |0.00   |50.08   |95.75   |0.11   |60.99   |94.95   |3.25   |**91.39**   |
> > |HiddenKiller   |95.00   |17.54   |75.32   |94.14   |0.00   |50.10   |95.41   |0.15   |51.88   |94.58   |14.40   |**74.57**   |
> >
> >
> > HSOL-Dataset:
> >
> >
> > |Methods   |ReTrain (CACC)   |ReTrain (LFR)   |ReTrain (PACC)   |GA (CACC)   |GA (LFR)   |GA (PACC)   |NPO (CACC)   |NPO (LFR)    |NPO (PACC)   |RGA (CACC)   |RGA (LFR)   |RGA (PACC)   |
> > |---|---|---|---|---|---|---|---|---|---|---|---|---|
> > |BadNets   |94.51   |5.97   |94.35   |94.77   |1.12   |62.64   |95.02   |6.11   |**93.71**   |93.32   |9.01   |93.31   |
> > |AddSent   |94.51   |5.92   |94.55   |91.89   |0.00   |50.03   |94.60   |0.62   |61.51   |93.25   |10.19   |**92.98**   |
> > |HiddenKiller   |94.51   |53.26   |72.65   |94.73   |0.00   |50.10   |94.74   |1.58   |52.74   |93.23   |46.93   |**74.02**   |
> >
> >
> > AG-Dataset:
> >
> >
> > |Methods   |ReTrain (CACC)   |ReTrain (LFR)   |ReTrain (PACC)   |GA (CACC)   |GA (LFR)   |GA (PACC)   |NPO (CACC)   |NPO (LFR)    |NPO (PACC)   |RGA (CACC)   |RGA (LFR)   |RGA (PACC)   |
> > |---|---|---|---|---|---|---|---|---|---|---|---|---|
> > |BadNets   |90.80   |11.65   |88.73   |90.57   |66.65   |25.03   |91.20   |8.98   |86.23   |89.73   |11.38   |**87.90**   |
> > |AddSent   |90.80   |12.22   |88.77   |90.13   |66.67   |25.00   |90.63   |7.64   |80.37   |89.53   |9.78   |**89.00**   |
> > |HiddenKiller   |90.80   |27.95   |73.80   |89.90   |66.49   |25.13   |90.47   |22.13   |67.73   |88.57   |18.75   |**78.10**   |
> >
> > As shown in the results from Qwen3-8B, GA leads to the worst PACC across different attack scenarios and datasets, while our method RGA consistently maintains PACC close to ReTrain across all datasets and attacks. This demonstrates that RGA effectively mitigates trigger shifting even on recent new LLMs.
> >
> > &nbsp;
> >
> > **W3**:
> > We conducted a new experiment on the [**Yahoo Answers Topics dataset**](https://www.kaggle.com/datasets/bhavikardeshna/yahoo-email-classification), which contains **10** categories (Society, Science, Health, Education, Computers, Sports, Business, Entertainment, Family, Politics). We randomly sample 10,000 texts per class for training (100,000 total) and 1,000 texts per class for testing (10,000 total). We set the target class as ''Society'' and set the poisoned ratio as 10% with the setting: learning rate = 2e-5, batch size = 16, 5-epoch poisoning, and 30-epoch unlearning. We evaluated ReTrain, GA, NPO, and RGA under the BadNets attack across BERT, DistilBert, Llama2-7B, and Qwen-8B. The experimental results are shown below.
> >
> > |   Methods  | Retrain (CACC) | Retrain (LFR) | Retrain (PACC) | GA (CACC) | GA (LFR) | GA (PACC) | NPO (CACC) | NPO (LFR) | NPO (PACC) | RGA (CACC) | RGA (LFR) | RGA (PACC) |
> > |:----------:|----------------|---------------|----------------|-----------|----------|-----------|------------|-----------|------------|------------|-----------|------------|
> > |    Bert    |      60.41     |     37.07     |      60.47     |   60.50   |   85.97  |   12.89   |    61.65   |   82.08   |    16.35   |    63.51   |   35.11   |    **62.76**   |
> > | Distilbert |      61.07     |     37.91     |      60.35     |   60.14   |   85.82  |   13.02   |    60.62   |   74.39   |    23.28   |    62.64   |   35.86   |    **62.44**   |
> > |  Llama2-7B |      65.60     |     32.78     |      65.11     |   65.19   |   85.08  |   13.79   |    67.15   |   74.56   |    23.26   |    67.68   |   30.48   |    **67.32**   |
> > |   Qwen-8B  |      66.33     |     32.64     |      66.25     |   65.98   |   86.12  |   12.72   |    66.61   |   68.03   |    28.99   |    65.31   |   32.82   |    **65.37**   |
> >
> > We have similar observations on this new 10-class dataset:
> >
> > 1. For all models, ReTrain has CACC $\approx$ PACC, indicating that the clean model is unaffected by the trigger.
> > 2. Under GA, the LFR increases to 86%, and the PACC correspondingly drops to 12–14%. This means that almost all samples are predicted as one dominant class, confirming a severe trigger shifting phenomenon also in a 10-class setting. Similarly, the PACC achieved by NPO also drops significantly, showing the issue of trigger shifting.
> > 3. Compared with GA and NPO, RGA achieves similar CACC and PACC compared with ReTrain.

---

> > > ### Author Response · Authors · 2025-11-21
> > >
> > > **Minor Weakness 1 & 2**
> > > Thank you for your suggestion. We will update our paper in the next version.
> > >
> > > &nbsp;
> > >
> > > **Q1**: Thank you for pointing this out. Table 6 is an ablation that compares DGA (term i + ii) with the full RGA (term i + ii + iii) at an early stage of unlearning (10 epochs). We would like to clarify that across datasets and attacks, RGA achieves lower ΔPACC than DGA in the majority of cases. This is because DGA behaves like a more aggressive variant of RGA without the regularization term that enforces the updated model to be close to the base model. Therefore, it can sometimes appear slightly better at early epochs, while RGA remains stable.
> > >
> > > **Q2**:
> > > Below shows the experimental results on the HSOL dataset and using BERT as the base model. RGA_a indicates the unlearning conducted on "actual poisoned examples" while RGA_d indicates the unlearning conducted on samples "detected by CUBE". Overall, using detected samples by CUBE for unlearning achieves similar performance compared with the one using actual poisoned samples.
> > >
> > >
> > > |              | RGA_a (CACC) | RGA_a (LFR) | RGA_a (ΔPACC) | RGA_d (CACC) | RGA_d (LFR) | RGA_d (ΔPACC) |
> > > |:------------:|--------------|-------------|---------------|--------------|-------------|---------------|
> > > |    BadNet    |     94.85    |     7.08    |      0.07     |     93.75    |     5.85    |      1.31     |
> > > |    AddSent   |     94.59    |     6.33    |      0.53     |     93.90    |     6.65    |      1.00     |
> > > | HiddenKiller |     94.65    |    44.86    |      0.28     |     93.10    |    45.08    |      0.42     |
> > >
> > > **Q3**: Our work focuses on textual backdoor unlearning. However, the mechanism we analyze is modality-agnostic: the proofs in Propositions 1 and 2 only assume a supervised classification model trained with cross-entropy on a poisoned subset $D_p$. Conceptually, the same trigger-shifting effect can therefore be observed in vision models if one applies vanilla GA-based unlearning on backdoored image classifiers.
> > >
> > >
> > > **Q4**: Our current experiments focus on classification-based tasks and systematically evaluating RGA on generation (e.g., refusal behavior or harmful content) is an exciting direction for future work. We will clarify this potential extension in the conclusion.

---

> > > > ### Comment · Reviewer_acST · 2025-11-25
> > > >
> > > > Thank you for the explanation. I increase my score.

---

### Official Review · Reviewer_6mL3 · 2025-10-29

**Soundness:** 4
**Presentation:** 4
**Contribution:** 3
**Rating:** 8
**Confidence:** 4

**Summary:**

This paper addresses the problem of removing backdoor effects from poisoned language models through a new method called Robust Gradient Ascent (RGA). The authors first identify and formalize the trigger-shifting problem (a situation in which the backdoor is not eliminated but merely moved into another class) that arises when applying unlearning approaches to remove the influence of backdoor attacks. To mitigate this, the proposed RGA introduces an adaptive penalty term that dynamically modulates the unlearning process to prevent divergence and preserve model utility. Experimental evaluations are conducted on three text classification datasets, three model architectures, and under three distinct backdoor attacks. The results demonstrate that RGA stabilizes the unlearning process and prevents trigger-shifting.

**Strengths:**

The paper is well written and organized. The motivation is clearly stated, the mathematical formulation is elegant, and the proofs are concise and well integrated into the appendix.

The methodology is conceptually simple yet effective. The adaptive penalty strategy in RGA is a neat and intuitive idea that provides theoretical grounding for stability and convergence while maintaining interpretability.

The experimental setup is comprehensive and convincing. The authors evaluate across multiple datasets, architectures, and attack types, providing a strong empirical validation.

Each research question is well defined, and the experimental section is systematically designed to address it.

**Weaknesses:**

**Multiclass setup.** The experimental evaluation is limited to two-class or, at most, four-class classification problems. The theoretical framework for addressing trigger shifting is well motivated in the binary case, where the gradient ascent direction can be clearly interpreted as moving away from one class and toward another. However, it remains unclear whether this analysis holds when extending to a larger number of classes, where the gradient effects may be distributed across multiple class directions, potentially reducing the impact or stability of the unlearning process. To solve this limitation, the authors could include an additional experiment on a synthetic dataset with an increasing number of classes, or on a more complex multi-class setting, to examine whether the dynamics of RGA generalize beyond the binary scenario.

**Assumption on access to poisoned samples**. A minor recommendation is related to the assumption that the model owner has access to the poisoned data samples ( D_p ). This assumption is explicitly stated early in the text but not reinforced in Section 5, where the methodology is presented in detail.

**Minor issues**:
- The abstract Figure 1 is visually appealing but not very clear at first glance. The figure could benefit from a simplified layout and clearer labels to better convey the main conceptual flow.

- The notation ( f_\theta(y|x) ) could be simplified to ( f_\theta(x) ) throughout the paper. Since the method does not explicitly model conditional distributions in the optimization, enforcing the conditional notation adds unnecessary complexity and may confuse the reader.

**Questions:**

How does RGA behave when the detected poisoned dataset ( D_p ) contains false positives or false negatives? Does the adaptive regularization remain stable or does it diverge under misidentified samples?

---

> ### Author Response · Authors · 2025-11-21
>
> Thank you for your valuable suggestion. We conducted a new experiment on the [**Yahoo Answers Topics dataset**](https://www.kaggle.com/datasets/bhavikardeshna/yahoo-email-classification), which contains **10** categories (Society, Science, Health, Education, Computers, Sports, Business, Entertainment, Family, Politics). We randomly sample 10,000 texts per class for training (100,000 total) and 1,000 texts per class for testing (10,000 total). We set the target class as ''Society'' and set the poisoned ratio as 10% with the setting: learning rate = 2e-5, batch size = 16, 5-epoch poisoning, and 30-epoch unlearning. We evaluated ReTrain, GA, NPO, and RGA under the BadNets attack across BERT, DistilBert, Llama2-7B, and Qwen-8B. The experimental results are shown below.
>
> |   Methods  | Retrain (CACC) | Retrain (LFR) | Retrain (PACC) | GA (CACC) | GA (LFR) | GA (PACC) | NPO (CACC) | NPO (LFR) | NPO (PACC) | RGA (CACC) | RGA (LFR) | RGA (PACC) |
> |:----------:|----------------|---------------|----------------|-----------|----------|-----------|------------|-----------|------------|------------|-----------|------------|
> |    Bert    |      60.41     |     37.07     |      60.47     |   60.50   |   85.97  |   12.89   |    61.65   |   82.08   |    16.35   |    63.51   |   35.11   |    **62.76**   |
> | Distilbert |      61.07     |     37.91     |      60.35     |   60.14   |   85.82  |   13.02   |    60.62   |   74.39   |    23.28   |    62.64   |   35.86   |    **62.44**   |
> |  Llama2-7B |      65.60     |     32.78     |      65.11     |   65.19   |   85.08  |   13.79   |    67.15   |   74.56   |    23.26   |    67.68   |   30.48   |    **67.32**   |
> |   Qwen-8B  |      66.33     |     32.64     |      66.25     |   65.98   |   86.12  |   12.72   |    66.61   |   68.03   |    28.99   |    65.31   |   32.82   |    **65.37**   |
>
> We have similar observations on this new 10-class dataset:
>
> 1. For all models, ReTrain has CACC $\approx$ PACC, indicating that the clean model is unaffected by the trigger.
> 2. Under GA, the LFR increases to 86%, and the PACC correspondingly drops to 12–14%. This means that almost all samples are predicted as one dominant class, confirming a severe trigger shifting phenomenon also in a 10-class setting. Similarly, the PACC achieved by NPO also drops significantly, showing the issue of trigger shifting.
> 3. Compared with GA and NPO, RGA achieves similar CACC and PACC compared with ReTrain.
>
> &nbsp;
>
> **Assumption on access to poisoned samples** and **Question**
>
> Because in realistic settings, the poisoned samples are typically unknown, in all our experiments we first apply CUBE, a standard backdoor sample detection method, to identify poisoned examples and then use RGA to unlearn the backdoor from the poisoned model. That is, all backdoor unlearning results reported in the paper are obtained following the detect-then-unlearn pipeline rather than assuming oracle access to poisoned data.
>
> Since CUBE is not perfect under various attack types, the detected poisoned set $D_p$ contains either false positives or false negatives, or both (shown in Table 5 in the Appendix). Nevertheless, our experiments show that RGA still successfully removes the backdoor and mitigates trigger shifting even under imperfect detection.

---

> > ### Author Response · Authors · 2025-11-21
> >
> > To further clarify the trigger shifting in multi-classification tasks, we report the confusion matrices for the Attack, ReTrain, GA, and RGA  on 10,000 test samples, using Llama2-7B as the base model. Due to the character limitation, we show the results from Attack and ReTrain in this comment, while the results from GA and RGA are in the next comment.
> >
> > **Confusion Matrix under Attack**:
> > |               | Society | Science | Health | Education | Computers | Sports | Business | Entertainment | Family | Politics |
> > |:-------------:|---------|---------|--------|-----------|-----------|--------|----------|---------------|--------|----------|
> > |    Society    |   983   |    4    |    3   |     3     |     0     |    1   |     0    |       3       |    2   |     1    |
> > |    Science    |   902   |    78   |   12   |     5     |     0     |    0   |     1    |       0       |    0   |     2    |
> > |     Health    |   940   |    5    |   49   |     1     |     0     |    0   |     2    |       0       |    3   |     0    |
> > |   Education   |   947   |    14   |    1   |     32    |     1     |    2   |     0    |       1       |    0   |     2    |
> > |   Computers   |   959   |    0    |    0   |     0     |     37    |    0   |     4    |       0       |    0   |     0    |
> > |     Sports    |   952   |    1    |    1   |     1     |     0     |   44   |     0    |       1       |    0   |     0    |
> > |    Business   |   947   |    1    |    0   |     2     |     4     |    2   |    37    |       2       |    2   |     3    |
> > | Entertainment |   959   |    0    |    1   |     2     |     1     |    1   |     1    |       33      |    1   |     1    |
> > |     Family    |   951   |    0    |    1   |     1     |     0     |    1   |     2    |       3       |   41   |     0    |
> > |    Politics   |   946   |    1    |    0   |     2     |     0     |    1   |     1    |       0       |    0   |    49    |
> >
> > As the targeted class is the ''Society'', almost all poisoned samples in other classes are classified as ''Society'' by poisoned Llama2-7B.
> >
> > **Confusion Matrix after ReTrain**:
> > |               | Society | Science | Health | Education | Computers | Sports | Business | Entertainment | Family | Politics |
> > |:-------------:|---------|---------|--------|-----------|-----------|--------|----------|---------------|--------|----------|
> > |    Society    |   461   |    35   |   27   |    119    |     5     |   20   |    130   |       65      |   74   |    64    |
> > |    Science    |    26   |   678   |   54   |    140    |     18    |   13   |    44    |       11      |    3   |    13    |
> > |     Health    |    18   |    38   |   741  |     42    |     2     |   17   |    64    |       16      |   50   |    12    |
> > |   Education   |    57   |   129   |   24   |    521    |     24    |   26   |    109   |       29      |   25   |    56    |
> > |   Computers   |    5    |    14   |    4   |     35    |    809    |    8   |    74    |       30      |    6   |    15    |
> > |     Sports    |    10   |    16   |   15   |     25    |     8     |   822  |    49    |       33      |    8   |    14    |
> > |    Business   |    58   |    36   |   43   |    112    |     38    |   25   |    511   |       60      |   59   |    58    |
> > | Entertainment |    58   |    17   |   16   |     53    |     38    |   25   |    96    |      613      |   48   |    36    |
> > |     Family    |    70   |    6    |   39   |     30    |     15    |   13   |    102   |       46      |   653  |    26    |
> > |    Politics   |    50   |    18   |   12   |     62    |     4     |   10   |    96    |       22      |   24   |    702   |
> >
> > After retraining, the model achieves a performance similar to a clean model.

---

> > > ### Author Response · Authors · 2025-11-21
> > >
> > > **Confusion Matrix after vanilla gradient accent (GA)**:
> > >
> > > |               | Society | Science | Health | Education | Computers | Sports | Business | Entertainment | Family | Politics |
> > > |:-------------:|---------|---------|--------|-----------|-----------|--------|----------|---------------|--------|----------|
> > > |    Society    |    36   |    3    |    5   |     5     |     0     |    0   |     0    |      949      |    1   |     1    |
> > > |    Science    |    2    |    72   |   12   |     13    |     0     |    0   |     0    |      901      |    0   |     0    |
> > > |     Health    |    3    |    4    |   46   |     3     |     0     |    1   |     2    |      938      |    3   |     0    |
> > > |   Education   |    6    |    7    |    1   |     36    |     1     |    1   |     1    |      945      |    0   |     2    |
> > > |   Computers   |    0    |    0    |    0   |     1     |     37    |    0   |     3    |      959      |    0   |     0    |
> > > |     Sports    |    0    |    1    |    1   |     1     |     0     |   44   |     0    |      953      |    0   |     0    |
> > > |    Business   |    4    |    0    |    1   |     3     |     4     |    2   |    37    |      945      |    1   |     3    |
> > > | Entertainment |    1    |    0    |    1   |     1     |     0     |    0   |     1    |      993      |    1   |     2    |
> > > |     Family    |    8    |    0    |    2   |     1     |     0     |    1   |     3    |      946      |   39   |     0    |
> > > |    Politics   |    4    |    2    |    0   |     7     |     0     |    1   |     2    |      945      |    0   |    39    |
> > >
> > >
> > > **Confusion Matrix after RGA**:
> > > |               | Society | Science | Health | Education | Computers | Sports | Business | Entertainment | Family | Politics |
> > > |:-------------:|:-------:|:-------:|:------:|:---------:|:---------:|:------:|:--------:|:-------------:|:------:|:--------:|
> > > |    Society    |   475   |   43    |   34   |    62     |     6     |   18   |    49    |      61       |  149   |   103    |
> > > |    Science    |    19   |   791   |   67   |    45     |    13     |    8   |    23    |      11       |   7    |    16    |
> > > |     Health    |    19   |   55    |  764   |    14     |     2     |   14   |    28    |      10       |  77    |    17    |
> > > |   Education   |    76   |   183   |   26   |   427     |    22     |   22   |    78    |      25       |  40    |   101    |
> > > |   Computers   |     4   |   18    |    4   |    14     |   836     |   12   |    69    |      17       |  10    |    16    |
> > > |     Sports    |    13   |   27    |   24   |     8     |    12     |  809   |    26    |      42       |  16    |    23    |
> > > |    Business   |    54   |   54    |   49   |    54     |    49     |   21   |   469    |      62       |  94    |    94    |
> > > | Entertainment |    41   |   23    |   21   |    21     |    36     |   41   |    58    |     619       |  94    |    46    |
> > > |     Family    |    39   |    7    |   33   |    14     |    15     |   17   |    47    |      37       |  764   |    27    |
> > > |    Politics   |    31   |   20    |   14   |    28     |     4     |   19   |    66    |      16       |   24   |   778    |
> > >
> > > We can observe that GA pushes all the poisoned samples to the class "Entertainment" while our method RGA can mitigate trigger shifting.

---

### Official Review · Reviewer_Ttp5 · 2025-10-30

**Soundness:** 3
**Presentation:** 2
**Contribution:** 3
**Rating:** 6
**Confidence:** 3

**Summary:**

This paper proposes Robust Gradient Ascent (RGA), a novel framework that enhances the stability and reliability of GA-based backdoor unlearning. Specifically, this paper shows that GA does not necessarily eliminate the backdoor effect but can instead redirect it to a new backdoor effect. To address this, this paper proposes RGA, which introduces a dynamic penalty mechanism that adaptively regulates the strength of GA during backdoor removal. Extensive experiments demonstrate that RGA effectively eliminates backdoors without trigger shifting, while preserving model utility, and offers a more reliable GA-based defense against backdoor attacks.

**Strengths:**

1.This paper makes a significant contribution by presenting the discovery of trigger shifting. Based on this, this paper proposes RGA, with its innovative dynamic penalty mechanism based on KL divergence, as a principled and effective solution.

2.This paper conducted comprehensive experiments across diverse datasets, models, and attack methods. The introduction of the PACC and ΔPACC metrics is a novel and essential contribution.

3.The paper is well written and clearly structured.

**Weaknesses:**

1.This paper does not discuss or discuss other potential functions (e.g., linear decay, step functions, or other divergence measures) to prove that the design of this dynamic penalty mechanism is optimal.

2.This paper does not discuss the time comparison with other baseline methods, except for retraining.

**Questions:**

Please refer to the weaknesses.

---

> ### Author Response · Authors · 2025-11-21
>
> **W1**: We appreciate this insightful comment. Our penalty term defined in Eq.7 in the paper is designed based on the following key points:
>
> 1. The weight $\lambda$ should depend on how far the **current model** has moved from the **poisoned state** in terms of prediction probability distribution on poisoned samples, rather than on the training step or epoch index (This is the reason we don't use step functions). The KL divergence between the current and poisoned predictions directly measures this quantity.
>
> 2. As soon as the current model no longer behaves like the poisoned model on triggered inputs, the penalty should rapidly shrink so that the unlearning process does not continue to push the poisoned loss to infinity. Therefore, the decay factor $\lambda$ should be close to 1 when the current unlearning model $\theta_{c^\*}$ remains similar to the poisoned model $\theta_p$ and should approach 0 when $\theta_{c^\*}$ diverges from $\theta_p$. An exponential decay provides a simple and fast-converging mechanism that satisfies this property, which is why we do not adopt a linear decay.
>
> 3. The exponential-KL guarantees $\lambda \in (0,1]$, is differentiable, and can be computed from a forward pass without additional backpropagation, keeping the computational cost close to vanilla GA. This makes RGA easy to plug into GA-based unlearning pipelines.
>
> &nbsp;
>
> **W2**: We have conducted additional experiments on Llama2-7B to show the effectiveness of RGA in terms of unlearning time (seconds). Overall, our approach has similar unlearning time to GA and NPO.
>
>
> | Methods | Retrain (BadNets) | Retrain (AddSent) | Retrain (HiddenKiller) | GA (BadNets) | GA (AddSent) | GA (HiddKiller) | NPO (BadNets) | NPO (AddSent) | NPO (HiddenKiller) | RGA (BadNets) | RGA (AddSent) | RGA (HiddenKiller) |
> |---------|-------------------|-------------------|------------------------|--------------|--------------|-----------------|---------------|---------------|--------------------|---------------|---------------|--------------------|
> | SST-2   | 403               | 397               | 396                    | 32           | 33           | 32              | 43            | 45            | 41                 | 39            | 38            | 38                 |
> | HSOL    | 335               | 334               | 334                    | 34           | 35           | 33              | 50            | 50            | 54                 | 48            | 47            | 51                 |
> | AG      | 477               | 476               | 476                    | 44           | 44           | 41              | 58            | 61            | 63                 | 52            | 53            | 52                 |

---

### Official Review · Reviewer_1F4J · 2025-11-01

**Soundness:** 2
**Presentation:** 4
**Contribution:** 4
**Rating:** 4
**Confidence:** 4

**Summary:**

The authors identify and systematically demonstrate the problem of trigger shifting in traditional gradient ascent, and propose robust gradient ascent algorithm to alleviate the phenomenon.

**Strengths:**

(1) The observation of trigger shifting is very novel

(2)  The proposed unlearning method that avoids trigger shifting is elegant

(3) The proposed evaluation metric for this observation is well-defined

(4) The motivation is well presented (especially in Figure 1)

**Weaknesses:**

(1) The observation of trigger shifting is valuable; however, your theoretical explanations do not seem comprehensive. For example, intuitively, the trigger shifting issue is less severe as the number of parameters of models increase, but this is not reflected in your formula.

(2) Following (1), robust gradient ascent is not necessarily the best approach to resolve the issue.

(3) More advanced/SOTA models, such as the Qwen3 families should be evaluated to justify whether your observation holds with models exposed to more training data, or have more parameters.

Minor issue: Table 2 is really hard to read

**Questions:**

See weakness.

---

> ### Author Response · Authors · 2025-11-21
>
> Thank you for your valuable suggestions. We have conducted additional experiments on **Qwen3-8B**. Following our original experimental setup (learning rate = 2e-5, batch size = 16, 10-epoch poisoning and 30-epoch unlearning), we evaluated ReTrain, GA, NPO, and RGA across all three datasets and all attacks using three consecutive random seeds. The experimental results are shown below:
>
> SST2-Dataset:
>
>
> |Methods   |ReTrain (CACC)   |ReTrain (LFR)   |ReTrain (PACC)   |GA (CACC)   |GA (LFR)   |GA (PACC)   |NPO (CACC)   |NPO (LFR)    |NPO (PACC)   |RGA (CACC)   |RGA (LFR)   |RGA (PACC)   |
> |---|---|---|---|---|---|---|---|---|---|---|---|---|
> |BadNets   |95.00   |7.02   |94.21   |94.89   |0.33   |62.22   |95.99   |2.23   |90.76   |94.16   |9.69   |**92.77**   |
> |AddSent   |95.00   |7.35   |92.79   |95.57   |0.00   |50.08   |95.75   |0.11   |60.99   |94.95   |3.25   |**91.39**   |
> |HiddenKiller   |95.00   |17.54   |75.32   |94.14   |0.00   |50.10   |95.41   |0.15   |51.88   |94.58   |14.40   |**74.57**   |
>
>
> HSOL-Dataset:
>
>
> |Methods   |ReTrain (CACC)   |ReTrain (LFR)   |ReTrain (PACC)   |GA (CACC)   |GA (LFR)   |GA (PACC)   |NPO (CACC)   |NPO (LFR)    |NPO (PACC)   |RGA (CACC)   |RGA (LFR)   |RGA (PACC)   |
> |---|---|---|---|---|---|---|---|---|---|---|---|---|
> |BadNets   |94.51   |5.97   |94.35   |94.77   |1.12   |62.64   |95.02   |6.11   |**93.71**   |93.32   |9.01   |93.31   |
> |AddSent   |94.51   |5.92   |94.55   |91.89   |0.00   |50.03   |94.60   |0.62   |61.51   |93.25   |10.19   |**92.98**   |
> |HiddenKiller   |94.51   |53.26   |72.65   |94.73   |0.00   |50.10   |94.74   |1.58   |52.74   |93.23   |46.93   |**74.02**   |
>
>
> AG-Dataset:
>
>
> |Methods   |ReTrain (CACC)   |ReTrain (LFR)   |ReTrain (PACC)   |GA (CACC)   |GA (LFR)   |GA (PACC)   |NPO (CACC)   |NPO (LFR)    |NPO (PACC)   |RGA (CACC)   |RGA (LFR)   |RGA (PACC)   |
> |---|---|---|---|---|---|---|---|---|---|---|---|---|
> |BadNets   |90.80   |11.65   |88.73   |90.57   |66.65   |25.03   |91.20   |8.98   |86.23   |89.73   |11.38   |**87.90**   |
> |AddSent   |90.80   |12.22   |88.77   |90.13   |66.67   |25.00   |90.63   |7.64   |80.37   |89.53   |9.78   |**89.00**   |
> |HiddenKiller   |90.80   |27.95   |73.80   |89.90   |66.49   |25.13   |90.47   |22.13   |67.73   |88.57   |18.75   |**78.10**   |
>
> Across SST-2, HSOL, and AG, we consistently observe:
>
> **A. GA produces severe trigger shifting on Qwen3-8B -- even worse than on Llama2-7B**
>
> For example, on AG News, under BadNets/AddSent/HiddenKiller, GA degrades sharply, collapsing to roughly 25% PACC on all poisoned samples, which means that all poisoned samples are shifted to a single class. This shows that trigger shifting persists regardless of parameter size, since Qwen3-8B is larger than Llama2-7B.
>
> Besides, our paper clarifies that trigger shifting is caused by the unbounded growth of GA loss, which optimizes the poisoned objective without a natural stopping criterion. The theoretical analysis on trigger shifting (Propositions 1 & 2) also demonstrates that trigger shifting is architecture-agnostic and parameter-size-agnostic. Therefore, increasing model size does not eliminate trigger shifting, as the GA unlearning process continues to increase the poisoned loss.
>
>
> **B. RGA remains robust on Qwen3-8B**
>
> As shown in the results from Qwen3-8B, GA leads to a worst PACC across different attack scenarios and datasets, while our method RGA consistently maintains PACC close to ReTrain across all datasets and attacks. This demonstrates that RGA effectively mitigates trigger shifting even on recent new LLMs.
>
> &nbsp;
>
> **About the concern: "robust gradient ascent is not necessarily the best approach to resolve the issue"**
>
> We would like to clarify that we do not claim RGA is the single "best" solution to resolve the trigger shifting. Instead, our contributions are: 1) Identify a novel phenomenon -- trigger shifting; 2) Provide the first theoretical analysis showing why GA inherently causes trigger shifting; 3) Introduce an effective method that stabilizes GA during backdoor unlearning. Across BERT, Distilbert, Llama2-7B, and now Qwen3-8B, our experiments consistently show that RGA reliably mitigates trigger shifting under diverse models, datasets, and attack settings.

---

### Meta-Review · Area_Chair_oeat · 2026-01-07

**Summary:**

The paper introduces the idea of trigger shifting and proposes the RGA framework with a KL-divergence–based dynamic penalty, which is technically sound but whose novelty over existing methods needs clearer positioning. The experimental evaluation is broad, yet some results lack depth and stronger justification, and the new PACC and ΔPACC metrics require better motivation and validation. While the paper is generally well organized, parts of the method and metric descriptions need clearer and tighter presentation. Overall, despite these weaknesses, the work contains sufficient novel ideas and technical substance to justify acceptance as a poster

**Reviewer Concerns:**

I believe most of the concerns below have been addressed. However, there are some minor improvements that the author should have made in the final version.  The design of the dynamic penalty lacks justification: alternative functions (e.g., linear decay, step schedules, or other divergence measures) are not explored, and the theoretical analysis of trigger shifting is incomplete—especially regarding model size and parameter scaling, which is not reflected in the formulation.

The methodological choice of robust gradient ascent is not convincingly motivated as the best solution, and the paper omits runtime comparisons with key baselines beyond retraining.

**Reviewer Scores:**

Initial assessment of all authors except one was positive. As far as I realized, the author who gave a low score raised his score after the rebuttal and if the discussion continued.

---

### Decision · Program_Chairs · 2026-01-26

Accept (Poster)